# Controllable strain-driven topological phase transition and dominant surface-state transport in HfTe$_5$

Jinyu Liu[1], Yinong Zhou [1], Sebastian Yepez Rodriguez[1], Matthew A. Delmont [2], Robert A. Welser[1], Triet Ho [2], Nicholas Sirica [3], Kaleb McClure[4], Paolo Vilmercati [4], Joseph W. Ziller[5], Norman Mannella [4], Javier D. Sanchez-Yamagishi [1], Michael T. Pettes [3], Ruqian Wu [1] & Luis A. Jauregui [1] ✉

The fine-tuning of topologically protected states in quantum materials holds great promise for novel electronic devices. However, there are limited methods that allow for the controlled and efficient modulation of the crystal lattice while simultaneously monitoring the changes in the electronic structure within a single sample. Here, we apply significant and controllable strain to high-quality HfTe$_5$ samples and perform electrical transport measurements to reveal the topological phase transition from a weak topological insulator phase to a strong topological insulator phase. After applying high strain to HfTe$_5$ and converting it into a strong topological insulator, we found that the resistivity of the sample increased by 190,500% and that the electronic transport was dominated by the topological surface states at cryogenic temperatures. Our results demonstrate the suitability of HfTe$_5$ as a material for engineering topological properties, with the potential to generalize this approach to study topological phase transitions in van der Waals materials and heterostructures.

The concept of topology in condensed matter physics has revolutionized our understanding of the electronic band structure of solid-state materials[1–9]. Topological phase transitions (TPTs), which are quantum phase transitions between states with different topological order, can be achieved by manipulating certain physical parameters[2,5,10–14]. As observed in several pioneering experiments, such transitions from trivial insulators to topological insulators (TI) have been realized by tuning the lattice parameters and/or spin-orbit coupling through element substitution[15–17]. While substitutional doping can introduce disorder into materials, the application of strain is a cleaner method for anisotropically tuning the lattice constants in bulk or microscopic device samples[18,19]. The significance of strain-driven TPTs in solid-state

systems is of paramount interest, as it offers the potential to unlock new phases of matter[20]. Strain manipulation provides the means to control pseudo-magnetic fields[21,22], locally adjust the Berry curvature[23,24], and develop quantum devices tailored for applications in quantum electronics and spintronics. To fully harness this potential, it is crucial to identify quantum materials that exhibit sensitivity to strain in their topological properties.

Since the prediction of their nontrivial band topology[25], transition metal pentatellurides such as ZrTe$_5$ and HfTe$_5$ have shown a range of intriguing physical properties including the chiral magnetic effect[26,27], the anomalous Hall effect[28], and the three-dimensional (3D) quantum Hall effect[29,30]. The pentatellurides crystallize in a layered

[1]Department of Physics and Astronomy, University of California, Irvine, CA 92697, USA. [2]Department of Mechanical and Aerospace Engineering, University of California, Irvine, CA 92697, USA. [3]Center for Integrated Nanotechnologies (CINT), Materials Physics and Applications Division, Los Alamos National Laboratory, Los Alamos, NM 87544, USA. [4]Department of Physics and Astronomy, The University of Tennessee, Knoxville, TN 37996, USA. [5]Department of Chemistry, University of California, Irvine, CA 92697, USA. ✉e-mail: lajaure1@uci.edu

orthorhombic structure with a space group of *cmcm*, as shown in Fig. 1a for the crystal structure of HfTe$_5$. As the building block of a two-dimensional (2D) layer, each HfTe$_3$ trigonal prism is formed by a Te-d dimer and an apical Te-a atom. The trigonal prisms with a one-dimensional (1D) characteristic along the *a*-axis are linked by parallel Te-z atoms with zig-zag chains along the *c*-axis, assembling a 2D HfTe$_5$ layer in the *ac* plane. The HfTe$_5$ layers are stacked together along the *b*-axis through van der Waals interactions to form a 3D layered crystal. The topological nature of as-grown samples has been debated, with different experiments finding that the pentatellurides can be either a weak topological insulator (WTI)[31–34], a strong topological insulator (STI)[27,35,36], or a Dirac semimetal (DSM)[26,37–39]. This inconsistency may be due to the pentatellurides ground state lying close to the phase boundary between a WTI and an STI phase[25]. First-principles calculations have shown that the topological character of the pentatellurides can change significantly with slight variations in the lattice parameters[25]. For 3D TIs, the STI phase exhibits topological surface states (TSS) on all of the surfaces, while a WTI phase hosts TSS only on certain surfaces. Surface electrons in an STI are topologically protected against backscattering, while those in a WTI phase are not necessarily protected[4]. A WTI and an STI have distinct topological characteristics and a transition between the two phases is not possible without closing and re-opening the bulk gap. For HfTe$_5$, in the WTI phase the TSS are predicted to exist only on the side surfaces, i.e. (100) and (001) planes, but not on the top and bottom surfaces along the (010) planes. The WTI phase exhibits similarities to 2D quantum spin Hall edge states that are stacked along the *b*-axis. HfTe$_5$, being at the phase boundary between a WTI and an STI phase, could be the ideal material candidate for studying TPTs driven by strain.

Recently, indications of a strain-tuned TPT from a WTI to an STI phase in ZrTe$_5$ were revealed from the transport properties of the bulk charge carriers, i.e., the manifestation of the chiral anomaly by negative longitudinal magnetoresistance (NLMR)[40] and the mass gap size extracted from the quantum oscillations[41]. In addition, also in ZrTe$_5$, the WTI state was observed by angle-resolved photoemission spectroscopy (ARPES) with the bulk gap tunable with external strain[33]. However, the defining transport evidence of an STI phase, the dominant contribution of the TSS upon the application of strain, is still missing. Moreover, less attention has been paid to HfTe$_5$ even though it has a larger spin-orbit coupling than ZrTe$_5$, which potentially could facilitate the formation of band inversion.

Here, we present a comprehensive study of the strain-driven TPT from a WTI to an STI phase in high-quality HfTe$_5$ samples from both in-depth first-principles calculations and extensive quantum transport experiments. First, by combining the Density Functional Theory (DFT) calculated phase diagram of HfTe$_5$ and the electrical transport measurements in a small strain range, we show our as-grown HfTe$_5$ samples are in a WTI phase. With the application of a relatively small strain (~ 0.3 %), we observe the closing of the band gap, showing dramatic changes in the temperature dependence of the resistivity. As strain is increased further, the band gap reopens converting the low-temperature behavior from initially metallic to insulating. With the largest amount of strain (~ 4.5%) applied, the resistivity saturates at cryogenic temperatures, hinting at a dominant TSS transport as observed in other well-studied 3D TIs[42–45]. In addition, under a perpendicular magnetic field (**B** // *b*-axis), the Shubnikov−de Haas (SdH) oscillations persist at the highest strain, displaying a nontrivial Berry phase of π at all strain levels. However, the associated Fermi surface shifts from a 3D nature at low strain levels to a 2D nature at higher strains, as revealed by both

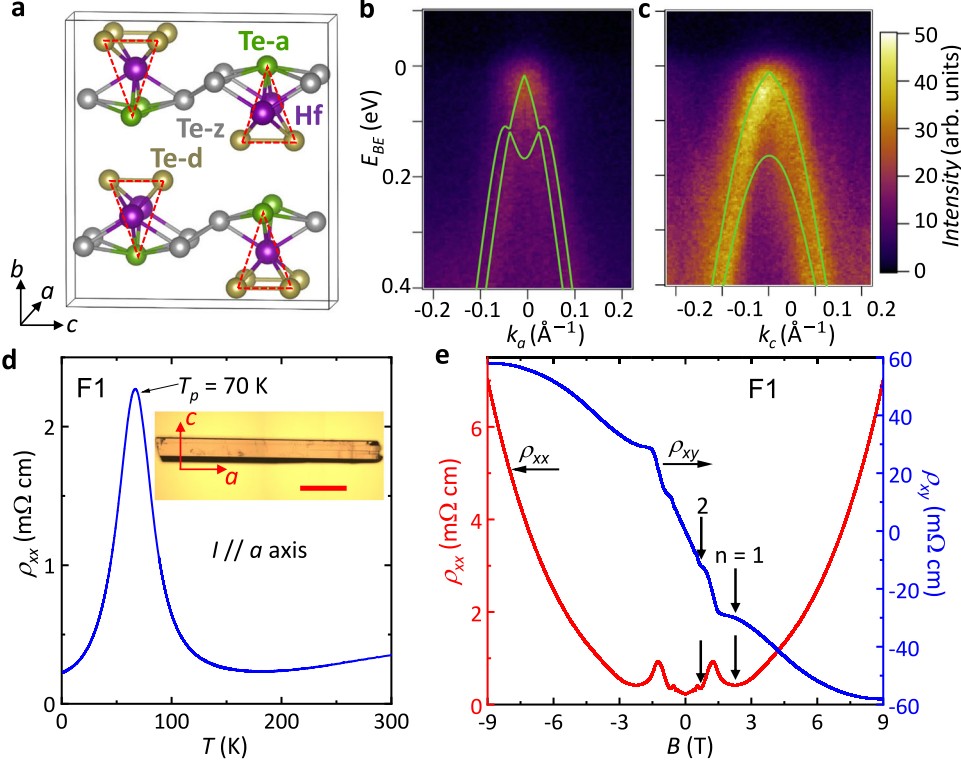

**Fig. 1 | Characterization of high-quality HfTe$_5$ single crystals. a** Crystal structure of HfTe$_5$. Te-d, Te-z, and Te-a represent Te atoms at dimer, zig-zag, and apical positions, respectively. The red dashed triangles depict the base of the Hf centered HfTe$_3$ prism. **b, c** ARPES results for the dispersions along X-Γ-X and Y-Γ-Y directions, respectively. The bright green lines are the band dispersions calculated by DFT, which agree well with the ARPES data. **d** Temperature (*T*) dependence of the resistivity ($\rho_{xx}$) for a free-standing HfTe$_5$ sample, F1, without any external strain applied. Inset: An optical image of a belt-like HfTe$_5$ single crystal, and the scale bar represents 0.2 mm. **e** Longitudinal ($\rho_{xx}$) and Hall ($\rho_{xy}$) resistivity of F1 plotted as a function of magnetic field (**B**) up to 9 T at *T* = 1.5 K. The arrows mark the La*n*dau level indices *n* = 1 and 2.

phase analysis and angular dependence measurements. Under a parallel magnetic field (**B** // *a*-axis), the SdH oscillations vanish, and the magnetoresistance becomes linear with the magnetic field at high strains. These electrical transport results, along with the DFT calculations, serve as unequivocal evidence of the emergence of a gapless TSS within the STI phase of HfTe$_5$, all resulting from the strain-induced topological phase transition.

## Results

We have synthesized belt-like HfTe$_5$ single crystals with a typical length of 1 cm by the chemical vapor transport (CVT) method (see Methods). The full lattice information of our samples, obtained from the refinement of single crystal diffraction is included in Section I of the Supplementary Information (SI). The crystallographic *a* and *c* axes are along the in-plane length and width directions respectively. The electronic structure is measured by ARPES. Measurements were carried out on an in-situ cleaved (010) surface, where high symmetry cuts along the XΓX ($k_a$) and YΓY ($k_c$) axes are shown in Fig. 1b, c, respectively. Here, at $T = 77$ K, prominent hole-like dispersion near the Brillouin zone center is observed in the vicinity of the Fermi level ($E_F < E_{BE} < 0.4$ eV), where $E_{BE}$ is the binding energy. A single Fermi surface pocket is detected at the $E_F$. The overlaid green lines correspond to the first-principles calculation results as discussed below and plotted here to show the agreement between ARPES experiments and calculations. While no photon energy dependence was performed in this study, the similarity of the measured electronic structure with literature suggests bands dispersing towards $E_F$ to be bulk-derived[34,46].

We have characterized the electrical properties of our as-grown HfTe$_5$ by measuring free-standing samples as described in Methods. For all the electrical transport experiments in this study, the current (*I*) is applied along the *a*-axis. A representative temperature dependence of the resistivity ($\rho_{xx}$) measured by the four-probe method in sample F1 is shown in Fig. 1d. As the sample is cooled down, $\rho_{xx}$ decreases slightly between $180$ K $< T < 300$ K. As the sample is cooled further, $\rho_{xx}$ increases drastically for $T < 180$ K until a peak in $\rho_{xx}$ is reached at $T_p \sim 70$ K. For $T < T_p$ the sample behaves as a metal (decreasing $\rho_{xx}$ with decreasing *T*). Our measured $T_p \sim 70$ K agrees well with previously studied HfTe$_5$ samples grown by CVT[30]. Recent ARPES experiments attributed $T_p$ to a temperature-induced Lifshitz transition, where the chemical potential gradually shifts from the valence band to the conduction band of the gapped Dirac cone as the temperature is reduced[46,47], with a hole-dominated transport for $T > T_p$ and electron-dominated transport for $T < T_p$, while the chemical potential is in the gap for $T_p$, as previously reported[30,48] and measured in our samples (Supplementary Fig. 3). One of the explanations of the Lifshitz transition is the reduced lattice constant with the reduced temperature, where the lattice constant *b* was shown to reduce by 0.3% when cooling down from 300 K to 4K[46]. Other plausible explanations are based on Dirac polarons[49] and thermodynamically induced carriers[50], but the origin is still under debate.

We have conducted magnetoresistance measurements on our as-grown samples. Specifically, the $\rho_{xx}$ and the transverse Hall resistivity ($\rho_{xy}$) of sample F1 at $T = 1.5$ K are presented as functions of the perpendicular magnetic field (**B**) in Fig. 1e. $\rho_{xx}$ displays clear SdH oscillations starting from $B = 0.2$ T. The SdH oscillations have a frequency of 0.97 T, corresponding to a small Fermi surface cross-sectional area (for **B** // *b*-axis) $\sim 0.922 \times 10^{-4}$ Å$^{-2}$, extracted by the Onsager relation. The $\rho_{xy}$ vs. **B** is linear for **B** $< 0.2$ T and displays quantum oscillations similar to those seen in $\rho_{xx}$. From the measured $\rho_{xy}$ vs. **B**, we extract an electron carrier density of $3.76 \times 10^{16}$ cm$^{-3}$ (extracted from a single band model) and electron mobility of 755,000 cm$^2$ V$^{-1}$ s$^{-1}$, which is among the highest mobilities reported in HfTe$_5$ samples. Beyond the Landau level, $n = 1$, the quantum oscillations of $\rho_{xy}$ develop an evident Hall plateau, while $\rho_{xx}$ reaches a minimum, demonstrating we observe the 3D quantum Hall effect in our samples, as observed in previous

studies[29,30]. These results are consistent with the electron transport being dominated by bulk massive Dirac fermions.

We further investigate the topological properties of HfTe$_5$ by first-principles calculations. The lattice constant and band gap are dramatically influenced by different exchange-correlation functionals. We test four different functionals: standard Perdew−Burke−Ernzerhof (PBE)[51], PBE-D3 with a van der Waals (vdW) correction[52], optB86b-vdW (a modified vdW-DF functional)[53], and the strongly constrained and appropriately normed (SCAN)[54] with the revised Vydrov−van Voorhis (rVV10)[55] (see Table S4 in the SI). The SCAN meta-generalized gradient approximation can accurately treat short- to intermediate-range vdW interactions. We conclude that SCAN with rvv10 vdW correction can better describe the TPT of the pentatellurides with Te−Te bonds in the range of 2.7−4 Å. The SCAN-rVV10 approach predicts HfTe$_5$ with no strain applied is in a WTI phase with the Dirac gap at the $\Gamma$ point and the $\mathbb{Z}_2$ indices (0;010) close to the phase transition point (see Sections V and VI of the SI).

Next, we study HfTe$_5$ under uniaxial strain along the *c*-axis. Figure 2a−c shows the band structure along X-$\Gamma$-Y high symmetry lines, where X′ is the point along X-$\Gamma$ direction. With a −1% strain applied along the *c*-axis the band gap at the $\Gamma$ point is opened (Fig. 2a). Under tensile strain the band gap closes (Fig. 2b) and reopens (Fig. 2c) with a band inversion between the *p* orbitals of Te-d atoms and Te-z atoms (labeled in Fig. 1a). At 1% tensile strain, the formation of the topological surface Dirac dispersion at the (010) top surface is evident in Fig. 2d. The tuning of the topological surface states and the bulk gap with strain is shown in the Supplementary Fig. 8. The evolution of the band gap at the $\Gamma$ point is depicted in Fig. 2e, where the gap closing point needs a minimal tensile strain, as indicated by the red dashed line. Based on these results, HfTe$_5$ can transition from a WTI to an STI phase by applying tensile strain along the *c*-axis. Similarly, the system can also transition from a WTI to an STI phase by applying compressive strain along the *a*-axis, as shown in Fig. 2f. At the $\Gamma$ point the band gap displays a minimum as a function of strain, across the phase transition (Supplementary Figs. 6 and 7). Our theoretical results confirm that HfTe$_5$ is in a WTI phase with no strain applied and a TPT to an STI phase is predicted with the application of tensile strain along the *c*-axis.

In order to experimentally confirm that our pristine HfTe$_5$ samples are in the WTI phase, we apply small strain along either the *a* or *c*-axis independently. Two different samples P1 and P2 are glued on the side wall of a piezo stack actuator with the *a*-axis perpendicular and parallel to the poling direction respectively, as shown in the inset of Fig. 2g. Note that, due to the negative thermal expansion coefficient of the piezo stack actuator, an inherent positive strain ($\epsilon_i$) can be applied to the samples when cooling down to low temperatures. A small strain ($\epsilon$) of up to +/- 0.13% can be continuously tuned and monitored (see Section VIII of the SI). Figure 2g shows the $\rho_{xx}$ *vs.* strain ($\epsilon$) for both of the samples measured simultaneously at $T = 70$ K. We choose $T = 70$ K because the chemical potential is at the band gap and the change of resistance is related to changes in the gap size. For sample P2 with strain applied along the *a*-axis ($\epsilon_a$), $\rho_{xx}$ increases monotonically with increasing $\epsilon_a$, indicating that $\epsilon_a$ favors a band gap-opening. While, for sample P1 with strain applied along the *c*-axis ($\epsilon_c$), $\rho_{xx}$ decreases monotonically with increasing $\epsilon_c$, indicating that $\epsilon_c$ favors a band gap-closing. Our ARPES measurements, electrical transport measurements, and the first-principles calculations agree that we are probing the bands around the $\Gamma$ point. Our experimental observation that the band gap closes with increasing tensile $\epsilon_c$ (or compressive $\epsilon_a$) agrees with the scenario that our HfTe$_5$ samples are intrinsically in the WTI phase[32−34,41,46]. However, our piezo stack actuator cannot be used to apply enough strain to observe a full gap closing (minimum of $\rho_{xx}$ vs. $\epsilon$) at $T = 70$ K. By cooling down the sample below $T < 70$ K we observe $\rho_{xx}$ can be reduced by increasing $\epsilon_c$ with a smaller effectivity (Supplementary Fig. 10). At $T = 16$ K positive $\epsilon_c$ (tensile strain) becomes ineffective at reducing $\rho_{xx}$ as $\rho_{xx}$ reaches a minimum. This minimum of $\rho_{xx}$

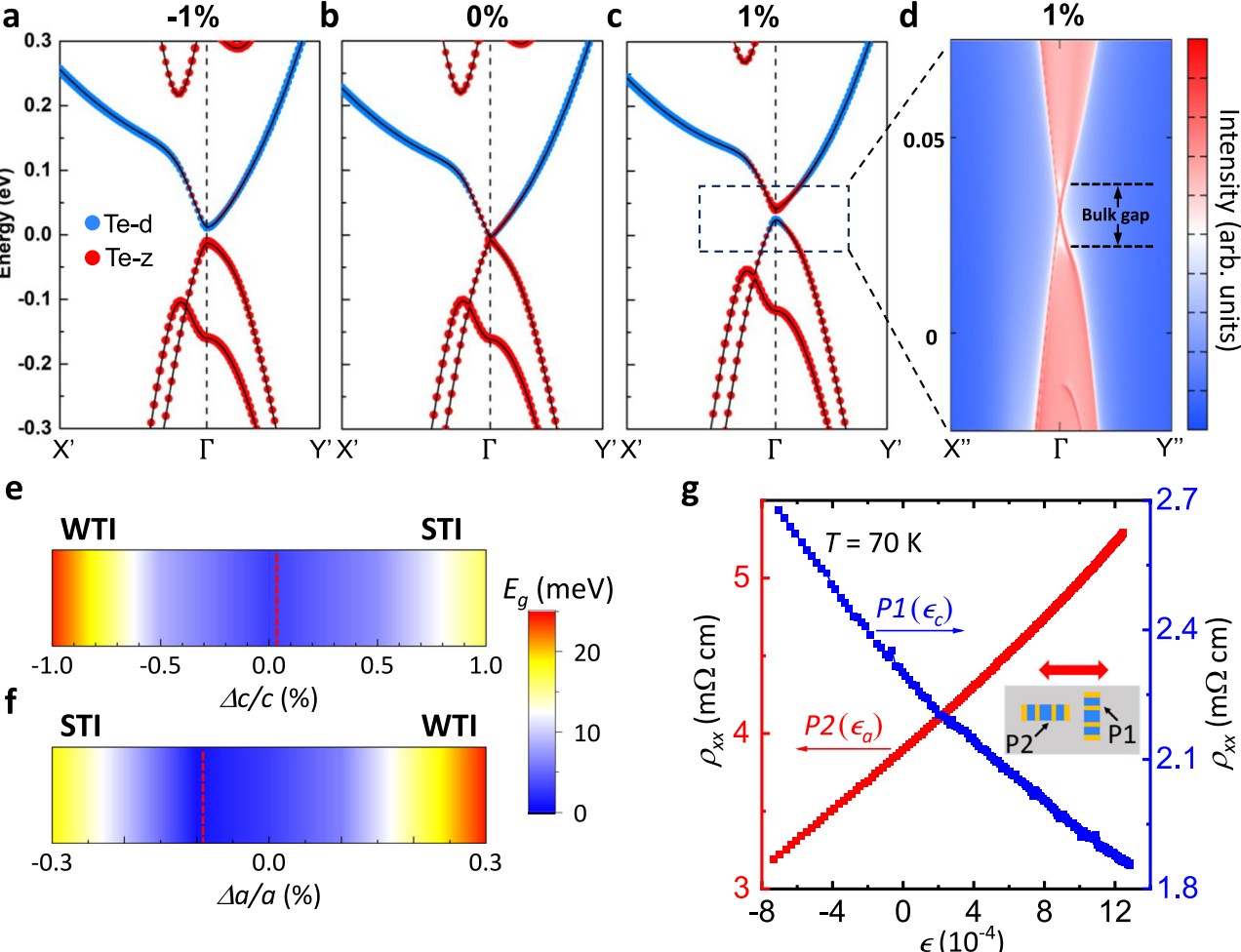

**Fig. 2 | Topological properties and topological phase transition under strain in HfTe₅.** **a**–**c** Band structure of HfTe₅ with different uniaxial strains applied along the *c*-axis (−1%, 0%, and 1% for a, b, and c, respectively). A band inversion occurs between the Te-d (blue dots) and Te-z (red dots) orbitals with tensile strain. We note that the electron doping due to possible Te deficiency of our as-grown samples is not reflected in the DFT calculations. **d** Enlarged view of the topological surface states spectrum of the top surface of HfTe₅ with 1% strain. Here, the sharp red lines represent the surface states, whereas the shaded regions show the spectral weight of projected bulk bands. Here, X' = 0.145 a*; X" = 0.07 a*; Y' = 0.375 c*;

Y" = 0.25 c*, where a* and c* are reciprocal lattice vectors. **e**, **f** The evolution of the band gap at Γ, with strain applied along the *c*-axis (Δ*c*/*c*) in (**e**) and along the *a*-axis (Δ*a*/*a*) in (**f**). The red lines represent the estimation of the gap closing point. The magnitude of strain needed to reach the weak topological insulator (WTI) or the strong topological insulator (STI) phase depends on the axis used to apply strain. **g** $\rho_{xx}$ as a function of strain for both samples P1 and P2 in a double y-axis plot. Inset: Schematic of sample configurations mounted on a single piezo-stack actuator for applying small strain along either *c* (sample P1) or *a* (sample P2) axis. The red double-headed arrow represents the poling direction of the piezo-stack actuator.

is related to the full closing of the band gap with the help of the inherent strain ($\epsilon_i \sim 0.4$ %) applied along the *c*-axis.

To achieve high and uniformly distributed strain on our samples, a different apparatus, the bending station, inspired by a similar one used for photoemission measurements elsewhere[56,57], is adapted here for electrical transport measurements, as seen in Fig. 3a. The details of our experimental design and finite element analysis (FEA) simulation are described in Methods. In short, an air-annealed titanium (Ti) beam is used as the sample substrate and secured against a sapphire bead by four screws, as seen in Fig. 3b i and ii. We use Ti because of the similar thermal expansion coefficient with HfTe₅. We apply strain by controlling the position of the four screws. FEA was employed to simulate the strain distributions for $\epsilon_y$ and $\epsilon_x$ along the length and width directions of the beam with a medium-high bending radius as depicted in Fig. 3b iii, iv respectively. A highly uniform strain distribution is ensured around the middle area on the top surface of the Ti beam. To drive our HfTe₅ samples from the WTI phase to the STI phase, compressive strain along the *a*-axis or tensile strain along the *c*-axis is required. Our measurement setup can prevent the sample from buckling, as the

sample is affixed at the center of the Ti beam's top surface with uniform strain distribution and we exclusively apply tensile strain along the *c*-axis when using the bending station, as shown in Fig. 3a. Conversely, applying large compressive strain along the *a*-axis would necessitate securing the sample to the bottom of the beam, resulting in non-uniform strain and potential buckling. We approximate the upper bound of strain applied to the sample by $\epsilon = t/(2R)$, where $\epsilon$ is the strain, *t* is the thickness of the beam ($t = 1$ mm), and *R* is the bending radius. We find that the extracted $\epsilon$ is in good agreement with $\epsilon_y$ from the FEA simulations.

The temperature dependence of $\rho_{xx}$ of sample B1 shows a remarkable evolution with strain as seen in Fig. 3c (for clarity Fig. 3d is plotted on a log-log scale). For our glued samples with no intentional strain applied at $\epsilon_0$, no significant difference is observed in the measured $\rho_{xx}$ vs. *T* when compared to the free-standing samples (sample F1). For *T* < 70 K, $\rho_{xx}$ is continuously decreasing, in contrast to sample P1 pasted on the piezo stack. Based on the radius analysis, we find an insignificant strain of $\epsilon_0 \sim 0.04$% which may be induced when securing the Ti beam against the sapphire bead by the four screws. As the beam

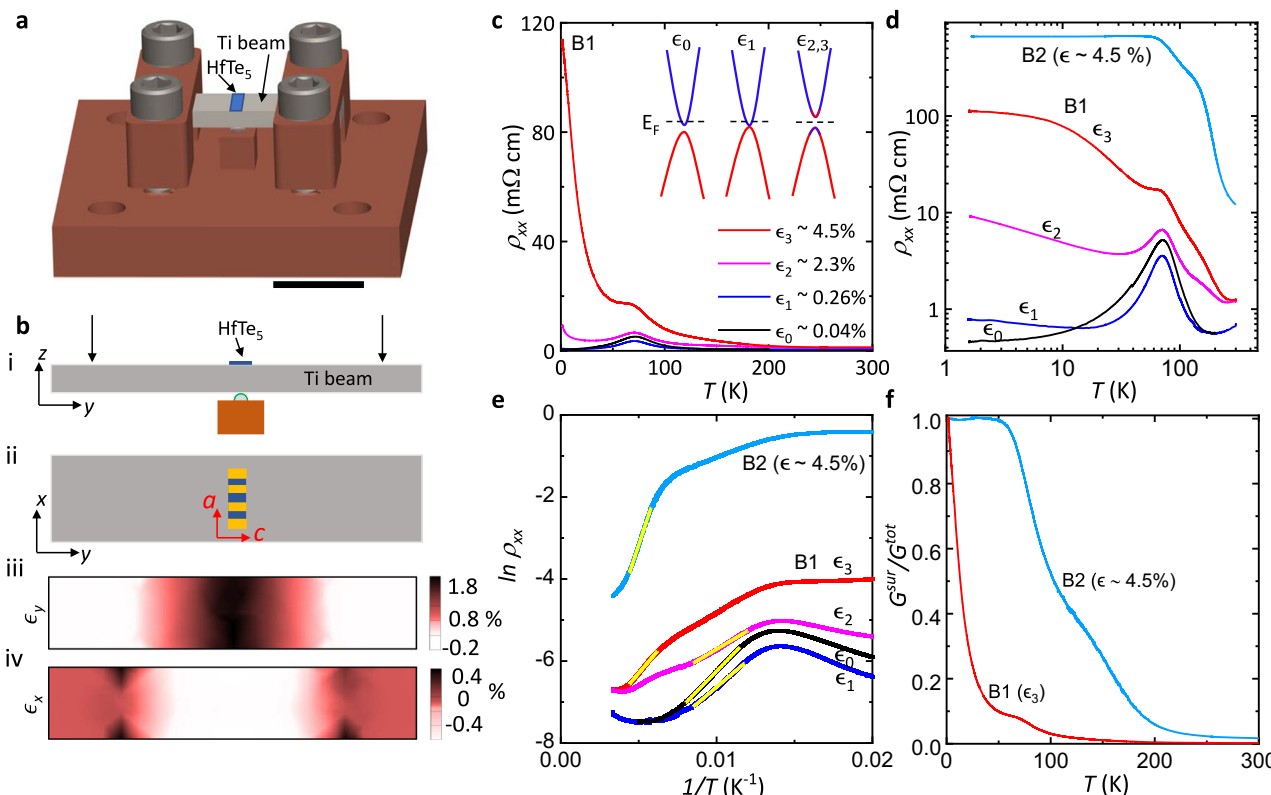

**Fig. 3 | Tuning electronic transport in HfTe₅ by large bending strain. a** A model of the bending station with a sample mounted at the center of the beam's top surface. The scale bar represents 1 cm. **b** (i) Schematic of how the beam is bent. The two black arrows represent the force load. (ii) Top view of the sample configuration relative to the beam for applying strain along the sample's $c$-axis. (iii) and (iv) show the strain distribution of $\epsilon_y$ and $\epsilon_x$ on the beam under a moderately high load, which results in a strain of $\epsilon_y = 2.2$ % near the sample area. **c** $\rho_{xx}$ as a function of temperature ($T$) for sample B1 at different strains and B2 at $\epsilon \sim 4.5\%$. Inset is a schematic of the Dirac bands around $\Gamma$ under different strain cases. **d** same plot as in (c) shown on a log-log scale for clarity. **e** $\ln(\rho_{xx})$ vs. $1/T$ at various strains, plotted for $T > 50$ K. The bright straight solid lines in e are the linear Arrhenius fit to extract the thermal activation energy. **f** The contribution of the surface state conduction ($G^{sur}$) to the total conduction ($G^{tot}$) vs. $T$, $G^{sur}$ is extracted from the fitting of sheet conductivity for sample B1 at $\epsilon_3$ and B2 at $\epsilon \sim 4.5\%$.

is further bent by tightening the four screws, the sample is strained more along the $c$-axis ($\epsilon_c$). By applying a small strain $\epsilon_1 \sim 0.26\%$, we notice a reduction of $\rho_{xx}$ by 32% at $T = 70$ K (when the chemical potential is at the band gap). This reduction of $\rho_{xx}$ with a small tensile strain $\epsilon_1$ agrees well with the result from sample P1 glued on a single piezo stack actuator (discussed above), where a 31% reduction of $\rho_{xx}$ is achieved with $\triangle\epsilon_c = 0.20\%$ at the same temperature (Supplementary Fig. 10b). As seen in Fig. 3d, for $20\,K < T < 150\,K$, $\rho_{xx}$ is much reduced when compared to the measurement at $\epsilon_0$. This may be caused by the reduced gap due to strain $\epsilon_1$. Such reduction of $\rho_{xx}$ caused by strain $\epsilon_1$ diminishes as the temperature is increased, and becomes negligible for $T > 200$ K. While below 20 K we observe a distinct temperature dependence compared to that of free-standing samples, we observe $\rho_{xx}$ increases with decreasing $T$. At $T = 20$ K, $\rho_{xx}$ shows a minimum, marking the closure of the band gap. For $T < 20$ K, the band gap increases or reopens by $\epsilon_1$ and low temperatures. A similar trend of resistivity upturn for $T < 30$ K was observed in sample P1 (Supplementary Fig. 10a), which experiences an effective strain of $\sim 0.475\%$ along the $c$-axis at base temperature (See detailed discussions in Section VIII of the SI). By increasing strain further, $\epsilon_2 \sim 2.3\%$, we observe a more obvious change of $\rho_{xx}$ vs. $T$. First, $\rho_{xx}$ vs. $T$ shows a more dominant insulating behavior. Indicating $\epsilon_2 \sim 2.3\%$ is enough to change the electronic structure significantly. As one can see from the high-temperature range, the metallic to semiconducting crossover temperature is $T \sim 240$ K, much higher than the crossover observed in the unstrained samples $\sim 200$ K. Second, the resistivity is dramatically enhanced, and $\rho_{xx}(\epsilon_2)/\rho_{xx}(\epsilon_0)$ becomes $\sim 2,000\%$ at $T = 1.5$ K. This

indicates that the band gap at low temperatures is reopening with the increased strain $\epsilon_2$. We increase strain further to $\epsilon_3 \sim 4.5\%$ and the measured $\rho_{xx}$ vs. $T$ shows an even more insulating behavior. Now, instead of showing crossovers between semiconducting and metallic behaviors, $\rho_{xx}$ increases monotonically as the temperature decreases. Compared with the $\epsilon_0$ conditions, the $\rho_{xx}$ at $T = 1.5$ K has increased by more than two orders of magnitude $(\rho_{xx}(\epsilon_3)/\rho_{xx}(\epsilon_0) = 24,200\%)$. Interestingly, $\rho_{xx}$ shows a saturation trend for $T < 10$ K, as seen in Fig. 3d. In another sample mounted on the bending station, B2, we observe an even more dramatic increase in $\rho_{xx}$ under the high strain $\epsilon \sim 4.5\%$ with $\rho_{xx}(\epsilon \sim 4.5\%)/\rho_{xx}(\epsilon_0)$ reaching 190,500%, at $T = 1.5$ K (See Fig. 3d and the Supplementary Fig. 15a, b). Consequently, a prominent saturation in $\rho_{xx}$ for $T < 50$ K is observed. The saturation in $\rho_{xx}$ observed in samples B1 and B2 at high strains of $\sim 4.5\%$ at low temperatures is reminiscent of the topological surface state dominant transport previously observed in 3D STIs[42–45].

The evolution of $\rho_{xx}$ vs. $T$ under different strains can be explained by the strain-induced TPT. From the Arrhenius analysis of $\rho_{xx}$ vs. $1/T$ shown in Fig. 3e, we evaluate the evolution of the bulk band gap. We selected temperatures exceeding 70 K to ensure that the estimate of the thermal activation gap remains unaffected by the Lifshitz transition. For sample B1, first for $80\,K < T < 120\,K$ at $\epsilon_0$, a thermal activation energy gap $\Delta \sim 33.6$ meV is extracted, consistent with the Dirac cone gap previously observed by ARPES in HfTe₅[34,46]. By applying strain $\epsilon_1 \sim 0.26\%$ the extracted gap is reduced to 27.7 meV, which confirms the closing of the band gap with strain as predicted by our DFT calculations. Under large strains $\epsilon_2 \sim 2.3\%$ and $\epsilon_3 \sim 4.5\%$, the insulating

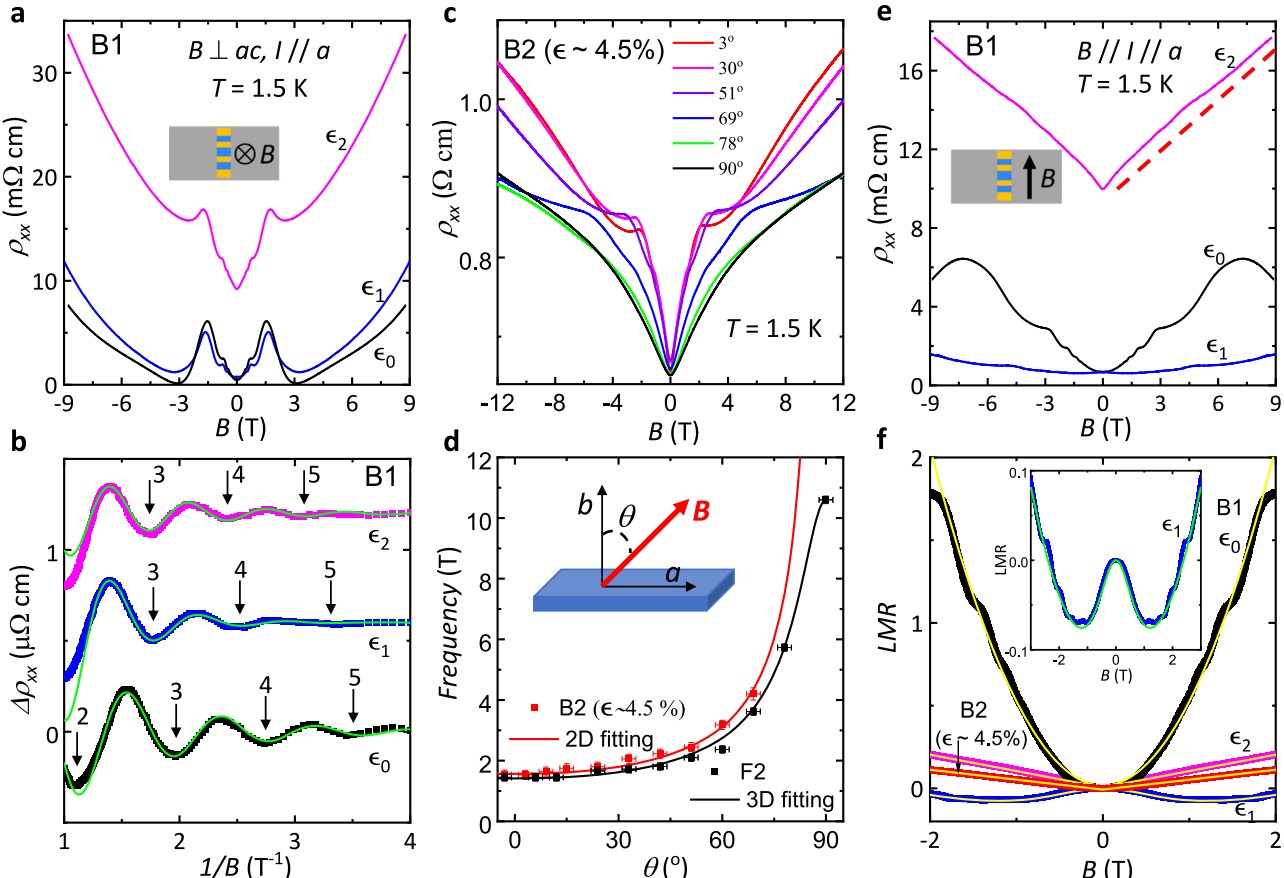

**Fig. 4 | Magnetoresistance in HfTe₅ under different strains. a** $\rho_{xx}$ as a function of perpendicular magnetic field (**B**) measured under different strains of B1. Inset: Schematic of the measurement setup. **b** Oscillatory part of the resistivity ($\Delta\rho_{xx}$) after background subtraction, plotted vs. $1/B$ in the field range from 0.25 T to 1 T. $\Delta\rho_{xx}$ is vertically shifted for clarity for strains $\epsilon_1$ and $\epsilon_2$. The bright solid green lines represent the LK fit (See Methods). The arrows mark the positions of integer Landau levels. **c** $\rho_{xx}$ vs. **B** measured at $T \sim 1.5$ K at different angles of B2 at $\epsilon \sim 4.5\%$. **d** Angular dependence of the oscillation frequency extracted by the fast Fourier transform (FFT) for samples F2 and B2 ($\epsilon \sim 4.5\%$). The solid lines represent the fitting of a 3D ellipsoid model $F(\theta) = F_\perp F_{//} / \sqrt{(F_{//}\cos\theta)^2 + (F_\perp\sin\theta)^2}$ (black) and a 2D model $F(\theta) = F_\perp / \cos\theta$ (red), where $F_\perp$ ($F_{//}$) is the frequency under a perpendicular (parallel) magnetic field. Inset shows the configuration between the sample and the magnetic field. The X error bars represent the instrumental error when controlling the gear baser rotator. The frequency error bars are caused by the uncertainty of the Fast Fourier Transform (FFT) peak positions. **e** $\rho_{xx}$ as a function of parallel magnetic field (**B**) at different strains of B1, measured with **B** // I (for this we rotate the sample and aligned the crystal $a$-axis to (**B**)). The red dashed line is a guide for the linear magnetic field dependence of $\rho_{xx}$ vs. **B** measured under strain $\epsilon_2$. Inset: Schematic of the measurement setup. **f** Longitudinal magnetoresistance (LMR) vs. **B** (up to 2 T) for B1 at different strains and B2 at $\epsilon \sim 4.5\%$. The bright solid lines represent the fittings with different equations, $LMR(\mathbf{B}) = \eta\mathbf{B}^2$ for $\epsilon_0$, $LMR(\mathbf{B}) = \frac{1}{\sigma(0) + \eta_1\mathbf{B}^2} - \frac{1}{\sigma(0)} + \eta_2\mathbf{B}^2$ for $\epsilon_1$, and $LMR(\mathbf{B}) = k|\mathbf{B}|$ for B1 at $\epsilon_2$ and B2 at $\epsilon \sim 4.5\%$. Where $\eta_1$, $\eta_2$, $\sigma(0)$, and $k$ are fitting parameters. Inset: Zoomed-in plot of LMR vs. **B** (up to 3 T) for strain $\epsilon_1$.

behavior of $\rho_{xx}$ starts at higher temperatures. For $170\,K < T < 250\,K$ and under $\epsilon_3$ conditions, we obtain $\Delta \sim 38.6$ meV. Similar strain-induced change in the thermal activation gap is observed in sample B2, with $\epsilon \sim 4.5\%$, a much-enhanced bulk gap $\Delta \sim 98.8$ meV is extracted. The increase of the extracted energy gap and the increase of the starting $T$ of the insulating behavior for the $\epsilon_2$ and $\epsilon_3$ cases in samples B1 and corroborated in sample B2 under 4.5% strain indicate a reopening of the band gap, driven by large strain. The saturation of the $\rho_{xx}$ observed in samples B1 and B2 for $T < 10$ K and $T < 50$ K respectively, can be well described by a model that incorporates thermally activated bulk conductance and metallic surface conductance[45], as illustrated in the Supplementary Fig. 15c. Detailed discussions about the fitting are given in the SI, section IX. From the fitting results, we estimate the contribution of the surface conduction to the total conduction is 100% at low temperatures and decays rapidly at high temperatures, as shown in Fig. 3f.

To further investigate the strain-induced change in the electronic structure, we have performed magnetoresistance measurements under different strains. Figure 4a shows $\rho_{xx}$ vs. **B** in sample B1 at various strains, including $\epsilon_0$, $\epsilon_1$, and $\epsilon_2$, under a perpendicular

magnetic field (**B**//$b$-axis) and at $T = 1.5$ K. Our sample shows strong SdH oscillations under all strain conditions, even for the $\epsilon_2$ case when the sample becomes very resistive. To study the SdH oscillations, we plot the oscillatory part ($\Delta\rho_{xx}$) after background subtraction in Fig. 4b. We observe the frequency of the SdH oscillations under strain $\epsilon_2$ is $1.4 \pm 0.2$ T, which is 16% larger than that of the unstrained $\epsilon_0$ case, $1.2 \pm 0.2$ T. The 2D carrier density extracted from SdH oscillations by $n_{2D} = F/\Phi_0$, (where $F$ is the oscillating frequency and $\Phi_0$ is the magnetic flux quantum) increases from $5.85 \times 10^{10}$ cm$^{-2}$ for $\epsilon_0$ to $6.77 \times 10^{10}$ cm$^{-2}$ for $\epsilon_2$. If we only consider it to be the bulk transport, such an increase in the carrier density is contradictory with the increased sample resistivity and bulk band gap by strain. One plausible explanation is that, under strain $\epsilon_2$, the TSS start to contribute to the total conduction and the probed SdH oscillations are from the TSS. This is confirmed in sample B2 at $\epsilon \sim 4.5\%$, where the SdH oscillations surprisingly persist even though the sample has become insulating at all temperatures and a bulk Fermi surface should not exist. The emergence of the surface state transport is in line with the topological phase transition from a WTI to an STI phase driven by strain and a strong topological surface state conduction is

expected at the STI phase, as depicted by our DFT calculations in Fig. 2d as well as Fig. S8 in the SI.

Figure 4b clearly displays a systematic shift of the SdH oscillations as strain is increased in sample B1. This phase shift could be related to the change in the oscillation frequency and also to an oscillation phase difference. We note two interesting facts about the oscillations in the high field range ($B > 1$ T). First, the oscillating amplitude is anomalously enhanced near the 1st Landau level ($n = 1$), which does not follow the Dingle damping term with the magnetic field in the Lifshitz−Kosevich (LK) formula (See Methods). Second, the interval between the last peak and valley is smaller in $1/B$ than those for $n > 2$. Both indicate the electronic bands are greatly influenced at a higher field ($B > 1$ T) when approaching the quantum limit, which is expected given the large Landé g-factor and the tiny band gap of the system. To better evaluate the Berry phase from the SdH oscillations, we consider the low-field oscillations ($B < 1$ T) and fit them with the LK formula. The Berry phase $\Phi_B$ is connected with the fitting phase factor $\gamma$ in the LK equation via $\frac{\Phi_B}{2\pi} = 1/2 - \gamma - \delta$, where $\delta$ is a phase factor associated with the Fermi surface dimensionality. For Dirac fermions in the 2D case, $\delta = 0$ and for the 3D case, $\delta = \pm 1/8$, where the sign depends on the carrier type and the extremal Fermi surface cross-section. In bulk ZrTe$_5$, $\gamma = 0.125$ was measured previously and it was consistent with a $\pi$ Berry's phase, demonstrating Dirac fermions in a 3D Fermi surface with $\delta = -1/8$[37,38,58]. Similarly, for our unstrained samples, the fitted phase factor is $\gamma = 0.12 \pm 0.01$, as shown in Fig. 4b, and for strain $\epsilon_1$ the fitted $\gamma = 0.13 \pm 0.02$. Indicating the charge carriers responsible for the SdH oscillations for the unstrained and strain $\epsilon_1$ cases are bulk Dirac carriers with a 3D Fermi surface. However, for strain $\epsilon_2$, the fitted phase factor changes significantly to $\gamma = -0.006 \pm 0.040$, which deviates strongly from the 3D case and supports the scenario of 2D Dirac fermions with $\Phi_B = \pi$ and $\delta = 0$. The change of the dimensional factor is further verified in sample B2 (see Supplementary Fig. 15d and Supplementary Table 5). Importantly, the angular dependence of the SdH oscillations demonstrates that it is indeed a 2D Fermi surface responsible for the quantum oscillations under high strains. Figure 4c presents the magnetoresistance results for the magnetic field at selected angles $\theta$, with $\theta$ being the polar angle between the sample's b-axis and the magnetic field orientation. For sample B2 at $\epsilon \sim 4.5\%$ the SdH oscillations can no longer be observed for $\theta > 70°$, and the oscillation frequency follows $F(\theta) = F(0)/\cos\theta$, which is a typical frequency-angular relation for a 2D Fermi surface. A detailed evolution of the SdH oscillations is depicted in Supplementary Fig. 16. In contrast, as illustrated in Supplementary Fig. 16a, for a free-standing sample (F2), SdH oscillations are observed at any angle and the measured angular dependence of the oscillation frequency can be explained by a 3D ellipsoid Fermi surface.

Additionally, we present longitudinal magnetoresistance results with the sample's a-axis aligned (parallel to the electrical current direction) to the magnetic field direction. Figure 4e shows $\rho_{xx}$ vs. **B** for sample B1 at different strains. At $\epsilon_0$, we observe strong SdH oscillations with a frequency of 5.35 T, which corresponds to the anisotropic 3D bulk Fermi surface as observed in previous studies[29,30]. With increasing strain to $\epsilon_1$, the apparent changes are as follows: first, the magnetoresistance at $B = 9$ T is much reduced, second, the SdH oscillations are weakened, and third, a negative magnetoresistance appears at low magnetic fields, as seen in the inset of Fig. 4f. By increasing strain to $\epsilon_2$, the SdH oscillations vanish and $\rho_{xx}$ vs. **B** becomes positive and linear up to $B = 9$ T (Fig. 4e). For clarity, we show the longitudinal magnetoresistance, $LMR(\mathbf{B}) = (\rho_{xx}(\mathbf{B}) - \rho_{xx}(\mathbf{B}=0))/\rho_{xx}(\mathbf{B}=0)$, with a magnetic field range from -2 T to 2 T in Fig. 4f. For the $\epsilon_0$ case, the LMR can be well fitted by a quadratic equation $LMR(\mathbf{B}) = \eta\mathbf{B}^2$, where $\eta$ is a fitting parameter. Under strain $\epsilon_1$, the LMR exhibits a competition between negative and positive components. Specifically, we observe that the LMR first decreases with an increasing **B**, reaching a minimum around $B \sim 1.2$ T, after which, it starts to increase. This suggests that under strain $\epsilon_1$, an additional scattering mechanism may be relevant (see

Methods). The negative LMR in the low field range can be roughly described by a positive magneto-conductance with a quadratic field dependence $\sigma_1(\mathbf{B}) \propto \mathbf{B}^2$, while the positive LMR above $B = 1.25$ T exhibits a quadratic field dependence $MR(\mathbf{B}) \propto \mathbf{B}^2$. Without considering the exact type of mechanism, according to Matthiessen's rule (See Methods)[59], the net LMR can be written as $LMR(\mathbf{B}) = \frac{1}{\sigma(0) + \eta_1\mathbf{B}^2} - \frac{1}{\sigma(0)} + \eta_2\mathbf{B}^2$, where $\eta_1$, $\eta_2$, and $\sigma(0)$ are fitting parameters. Our data can be well fitted with this equation for **B** up to 3 T, as shown in Fig. 4f and the inset. The negative LMR up to -7% probed near $\mathbf{B} = 0$ may be attributed to the intrinsic chiral anomaly effect, as reported in previous studies on the pentatellurides[26,60], especially in the DSM phase which is in good agreement with our assessment that the band gap is closed under strain $\epsilon_1$. By increasing strain to $\epsilon_2$, the LMR can be well fitted by a linear equation $LMR(\mathbf{B}) = k|\mathbf{B}|$ (as shown in Fig. 4f for $B < 2$ T or Fig. 4e for **B** up to 9 T), where $k$ is a fitting parameter. In sample B2 with $\epsilon \sim 4.5\%$, the LMR seen in Fig. 4c at $\theta = 90°$ displays a linear magnetic field dependence for $B < 3$ T but develops to a sub-linear trend for $B > 3$ T. The low field range is linearly fitted and presented in Fig. 4f. The linear LMR we observe at high strains may be related to the spin polarization of the helical TSS, previously observed in other STI materials[61], in agreement with our assessment of the reopening of the band gap and the dominance of the TSS at high strains.

## Discussion

The strain-induced TPT observed in our experiments can be interpreted as follows. By applying tensile strain along the c-axis of HfTe$_5$, at first, the bulk gap is reduced until it is fully closed, and eventually, the band gap reopens with increasing strain. Once the gap reopens, the TSS are formed. In other words, increasing strain causes a topological phase transition from the WTI phase to a Dirac semimetal phase, and finally to an STI phase with non-trivial surface states contributing to the electronic transport. Our first-principles calculations are in good agreement and show that the bulk gap is initially reduced and then increased when increasing tensile strain along the c-axis in HfTe$_5$.

As far as we know, indications of in-situ strain-driven TPTs have been observed only in TaSe$_3$ and ZrTe$_5$ samples by ARPES[33,57,62] and electronic transport[40,41]. In TaSe$_3$, a strain-induced TPT from an STI phase to a trivial semimetal phase was observed by ARPES[57,62]. However, due to the dominant bulk band at the Fermi level[57,62], the appearance or disappearance of the TSS would be difficult to detect from transport measurements. Thus, quantum transport evidence of the TPT in TaSe$_3$ has not been reported yet. In ZrTe$_5$, an ARPES study observed the closing of the gap by applying strain along the a-axis[33], with the maximum compressive strain ($\epsilon_a \sim -0.3\%$) they could apply. The band gap was closed, but the STI phase was not reached in their study. There have been two other electronic transport studies focusing on transport signatures of the strain-tuned TPTs in ZrTe$_5$[40,41]. However, the samples in those studies were dominated by bulk carriers, and no transport signatures of the TSS associated with the STI phase were observed. In previous strain studies, the observed gap changes were small, and the manifestation of bulk insulating states (resulting in the reopening of the gap) required to confirm the TPT to an STI phase was absent. This absence was likely attributed to the limited compressive strain ($\epsilon_a < 1\%$) that could be applied before inducing buckling in the sample. On the other hand, in this study, instead of applying compressive strain along the crystallographic a-axis, we apply tensile strain along the c-axis with a custom-built bending apparatus, as shown in Fig. 3a, which allows the application of a much larger strain. Compared with that of the unstrained $\epsilon_0$ case, the resistivity at the base temperature increases by more than three orders of magnitude with the highest strain applied ($\sim 4.5\%$). By driving the system deeper into the STI phase, its temperature dependence of resistivity exhibits a semiconducting-like behavior from high temperatures and displays a remarkable saturation at low temperatures. This behavior can be well

fitted using a model considering both surface and bulk conductions[45], which results in a 100% surface conduction for $T < 50$ K ($T < 10$ K) for sample B2 (sample B1) under 4.5% strain[42–45]. The dramatic changes in resistivity with the application of strain reveal the evolution in the bulk gap across the TPT driven by strain. The remaining finite conductance with the gapped bulk state serves as a strong transport evidence of the exposed TSS.

Our magnetoresistance findings at various strains can be comprehensively explained by the dominance of bulk transport at low strains and the prevalence of surface-state transport at high strains. Under a perpendicular magnetic field (**B** // *b*-axis), the SdH oscillations persist even at high strains when the low-temperature resistivity behaves as an insulator. It is well known that such SdH oscillations can appear in 3D topological insulators without a bulk Fermi surface because of the 2D TSS[43,45]. While we also note that some Kondo insulators, such as $SmB_6$[63] and $YbB_{12}$[64] show similar quantum oscillations, whose origin remains a topic of debate, with proposed explanations ranging from topological surface states[65] to bulk charge-neutral quasiparticles[63,66]. Nonetheless, our observed quantum oscillations in $HfTe_5$ under high strains are most likely attributed to the TSS, as they exhibit distinct characteristics from those observed in Kondo insulators, such as a light cyclotron effective mass (See the Supplementary Fig. 14) and the 2D nature of the corresponding Fermi surface, as evidenced by our angular dependent measurements. Moreover, a detailed analysis of the oscillating phase shift, $\gamma = 1/2 - \frac{\Phi_B}{2\pi} - \delta$, corroborates that the nature of the SdH oscillations changes with strain. At zero strain $\epsilon_0$, it results in a phase shift $\gamma \sim 1/8$, which supports a 3D Fermi surface of bulk Dirac electrons with non-trivial $\pi$ Berry's phase and a dimensional phase factor $\delta = -1/8$. However, at high strains, the resulting phase shift $\gamma \sim 0$ signifies the persistence of the $\pi$ Berry phase coupled with a dimensional phase factor $\delta = 0$, as the case for 2D Dirac fermions. This aligns with the presence of non-trivial TSS once the sample enters the STI phase. An alternative explanation of our SdH analysis suggests that the system undergoes a Lifshitz transition under strain, causing the bulk Fermi surface to shift from a closed 3D ellipsoid-like shape to an open 2D cylindrical configuration with open orbits in the out-of-plane direction. However, this might not be expected as the interlayer interaction should be stronger, and the momentum transfer should be favored by the compressive strain along the interlayer direction (*b*-axis) under the in-plane tensile strain. In addition, the existence of a bulk 2D cylindrical Fermi surface contradicts our observation of a dramatic increase in $\rho_{xx}$ and the insulator-like temperature-dependent behavior observed in our samples.

Under a parallel magnetic field (**B** // *a*-axis), the $\mathbf{B}^2$ dependent LMR and the SdH oscillations can be interpreted well by effective mass anisotropy in the classic limit of the bulk 3D Fermi surface and the quantization of electron orbits as it approaches the quantum limit[67]. With a small strain applied, $\epsilon_1$, we observed a negative LMR which may be due to the chiral anomaly. The chiral anomaly is an imbalanced chirality and has been observed when the magnetic field is parallel to the electrical current in gapless or slightly gapped DSM state[60,68] such as $ZrTe_5$[26,40,58]. This supports the hypothesis of the closing of the bulk gap with strain $\epsilon_1$ in $HfTe_5$. At higher strains beyond $\epsilon_2$, the SdH oscillations and the negative LMR vanish. Instead, we observe a linear LMR without SdH oscillations. This finding agrees well with the formation of a 2D Fermi surface of Dirac electrons on the top and bottom surfaces with the chemical potential located within or in the vicinity of the bulk gap. The appearance of the compelling linear LMR may be related to the coupling between helically polarized spins of the TSS and the magnetic field, as previously reported in 3D STIs[68], which deserves further investigation.

In summary, we have performed a thorough study on the application of controllable large strain in high-quality $HfTe_5$ samples. We perform DFT calculations and measure the electronic transport properties under different strains. The electronic band structure calculations accurately confirm that the $HfTe_5$ ground state is located at the boundary between a WTI and an STI phase. By using a small strain, we prove the $HfTe_5$ samples are initially in the WTI phase. By applying large strain, we observe a strain-driven TPT from the initial WTI phase to a closing (DSM phase) and reopening of the band gap, and finally an STI phase. With a strain of ~4.5%, the sample resistivity at $T = 1.5$ K can be increased by more than three orders of magnitude. We measured signatures of the topological surface-state-dominated electronic transport in the STI phase of $HfTe_5$ from the magnetoresistance with both perpendicular and parallel magnetic fields. Our results demonstrate that $HfTe_5$ is an ideal prototype material for studying strain-driven topological quantum phenomena and has the potential to be used in strain-tuned topological quantum devices.

## Methods

### Crystal growth and structural characterization

The $HfTe_5$ single crystals in this study are grown by the chemical vapor transport (CVT) method. Hf pieces and Te lumps are mixed in stoichiometric ratios before being loaded into a quartz tube (inner diameter 14 mm). A small amount of $I_2$ (100 mg) is added as the transport agent. After being sealed under vacuum, the quartz tube with a length of ~16 cm is placed in a horizontal furnace and the temperature gradient is set to be 510 ℃ and 460 ℃. Belt-like single crystals up to 1 cm in length are accessible after 1 month of growth. A piece of as-grown crystal was isolated and cut into 1/3 of the length for the single crystal X-ray diffraction measurement. Care was taken while cutting the crystal to minimize straining the crystal edge. The crystal structure is confirmed to be orthorhombic with a space group of *Cmcm*.

### Transport measurements

The thickness of our transport measurement samples ranges from 50 to 110 mm, specifically $55 \pm 2$ μm for sample B1, and $67 \pm 2$ μm for sample B2. The electrical measurement is done with the four-probe method. To ensure good contact between electrodes and the sample, 10 nm Cr and 90 nm Au are evaporated on freshly cleaved $HfTe_5$ single crystals under homemade shadow masks in a high vacuum e-beam evaporator (Angstrom Engineering Inc.). Two-component H20E silver epoxy (Epoxy Technology, Inc.) is used to attach 50μm thick Pd-coated copper wires onto the sample and baked at 90 ℃ for 1 h in the inert gas atmosphere. The prepared samples for electrical transport measurements are then either measured as free-standing samples or pasted on the Ti beam for measurements under bending strain. The samples are cooled down with a helium gas environment in a cryostat equipped with a cryogen-free superconducting magnet (Cryomagnetics, Inc). The temperature is monitored with a Cernox temperature sensor integrated into the sample platform. The electrical measurements are performed by using SR830 lock-in amplifiers (Stanford Research Systems, Inc.). An AC current, typically, of 100 μA with a frequency of 17.777 Hz is applied along the samples' *a*-axis.

### Angle-resolved photoemission spectroscopy (ARPES) measurements

Laboratory-based ARPES experiments were performed on $HfTe_5$ single crystals cleaved in situ at $T = 77$ K in a base pressure $P < 1 \times 10^{-10}$ Torr. Photoelectrons were collected over a 30° solid angle following He $I_\alpha$ ($h\nu = 21.2$ eV) excitation using a Scienta R4000 electron energy analyzer. The total energy resolution was $\Delta E < 12$ meV, while an instrumental angular resolution of $\pm$ 0.5° gives rise to a total momentum resolution

$\Delta$ k < 0.02 Å$^{-1}$ for the photon energy used in these experiments. Calibration of the spectrometer work function was performed on polycrystalline gold at $T = 77$ K ($\phi = 4.345$ eV), providing an absolute reference for the Fermi level (E$_F$) as is necessary given the insulating nature of HfTe$_5$ measured from transport in this temperature range.

## DFT Calculations

Our first-principles calculations are performed with the projector-augmented wave pseudopotentials[69] using Vienna Ab initio Simulation Package[70] code. We have compared four different exchange-correlation functionals: standard Perdew–Burke–Ernzerhof (PBE)[51], PBE-D3 with a van der Waals (vdW) correction[52], optB86b-vdW (a modified vdW-DF functional)[53], and the strongly constrained and appropriately normed (SCAN)[54] with the revised Vydrov–van Voorhis (rVV10)[55]. An energy cutoff of 450 eV and an $8 \times 8 \times 4$ Monkhorst–Pack k-point grid is used[71]. The structure is optimized until the atomic forces are <0.01 eV/Å. The maximally-localized Wannier functions of HfTe$_5$ are fitted based on the Te-p orbitals by the Wannier90 code[72] and then the topological properties are calculated by the WannierTools[73].

When applying strain along the c-axis (or a-axis), the lattice constants a (or c) and b are reduced correspondingly. The Poisson's ratios are calculated from the energy minimization from DFT calculations, shown as follows with Eq. (1) for uniaxial strain along the c-axis and Eq. (2) for uniaxial strain along the a-axis:

$$\begin{cases} \frac{\Delta a}{a} = -0.29 \frac{\Delta c}{c} \\ \frac{\Delta b}{b} = -0.12 \frac{\Delta c}{c} \end{cases} \tag{1}$$

$$\begin{cases} \frac{\Delta c}{c} = -0.29 \frac{\Delta a}{a} \\ \frac{\Delta b}{b} = -0.05 \frac{\Delta a}{a} \end{cases} \tag{2}$$

## Application of bending strain

An air-annealed titanium beam is placed under two brackets with its center supported by a sapphire bead (Fig. 3b i and ii). As the brackets are driven downward by tightening the screws, the beam will be bent, creating tensile strain along the length direction (defined as x axis) on the top surface of the beam. Such a strain can be well approximated by $\epsilon = t/(2R)$, where $\epsilon$ is the strain, t is the thickness of the beam and R is the bending radius (See Section IX of the SI). The sample is pasted along the middle line of the beam top surface by insulating EP29LPSP two-component epoxy (Master Bond Inc). The baking process of the epoxy is done inside an Ar-filled glove box by following the instructions given by Master Bond. The sample's a-axis is aligned perpendicular to the x axis. Thus, as the beam is bent, a tensile strain along the c-axis and a compressive strain along the a-axis will be applied to the sample. The thin insulating oxide layer on the top surface of the beam produced by the air-annealing process, together with the insulating epoxy, enables us to do electrical transport measurements of samples under strain. A finite element analysis (FEA) simulation of the strain distribution on the beam along the x and y axes and on the sample's top surface was performed using the Stress Analysis environment in Autodesk Inventor 2023 and COMSOL Multiphysics software. The beam's dimensions are set to 2 mm × 10 mm × 1 mm (width, length, and thickness), and the Young's modulus and Poisson's ratio are set to 103 GPa and 0.36, respectively. While the Young's modulus of the sapphire bead was set to 345 GPa. Regarding the mesh construction of the assembly, the average element size was 10% of the model size (part-based measure was used) and the minimum element size was 20% of the average value, resulting in a total of 10378 elements and 17302 nodes in the simulation. A force of 290 N was applied downward to each bracket to bend the beam resulting in a 2.3% longitudinal strain at its center, which agrees with the value calculated from measuring the bending radius R.

## Quantum oscillations fitting with the Lifshitz–Kosevich (LK) formula

Lifshitz–Kosevich (LK) formula:

$$\Delta\rho \propto \mathbf{B}^\lambda R_T R_D R_S \cos\left[2\pi\left(\frac{F}{\mathbf{B}} + \gamma\right)\right] \tag{3}$$

Where $R_T = \alpha T\mu/[B\sinh(\alpha T\mu/\mathbf{B})]$ is the thermal damping term, $R_D = \exp(-\alpha T_D\mu/\mathbf{B})$ is the Dingle damping factor with $T_D$ being the Dingle temperature, and $R_S = \cos(\pi g\mu/2)$ is the spin damping factor. $\mu = \frac{m^*}{m_e}$ is defined as the ratio between cyclotron effective mass m* and the free electron mass $m_e$. $\alpha = 2\pi^2 k_B m_e/(\hbar e)$ is a constant, where $k_B$ is the Boltzmann constant and $\hbar$ is the Planck constant. g is the Landé g-factor. $\lambda$ is a factor depending on the dimensionality (1/2 and 0 for 3D and 2D cases respectively). $\gamma$ is a crucial fitting parameter of the oscillating phase, encapsulating information about the Berry phase and a dimensional phase factor, as discussed in the main text.

## Matthiessen's rule

For a single type of carrier, according to Matthiessen's rule,

$$\frac{1}{\tau} = \frac{1}{\tau_1} + \frac{1}{\tau_2} + \ldots \tag{4}$$

where $\tau$ is the total mean free time and $\tau_i$ is the mean free time caused by the $i^{th}$ scattering mechanism. As a result, the net resistivity can be written as the sum of the resistivity introduced by the $i^{th}$ scattering mechanism, i.e. $\rho(\mathbf{B}) = \rho_1(\mathbf{B}) + \rho_2(\mathbf{B}) + \ldots$. Different from the additive of conductivity for systems with different conduction channels or different types of charge carriers, here the resistivity caused by different scattering mechanisms experienced by the same carriers is additive. In our case, one scattering mechanism causes positive magneto-conductance, $\rho_1(\mathbf{B}) = 1/(\sigma_1(0) + \beta_1 \mathbf{B}^2)$, and another scattering mechanism causes positive magnetoresistance, $\rho_2(\mathbf{B}) = \rho_2(0) + \beta_2 \mathbf{B}^2$, where $\sigma_1(0)$, $\rho_2(0)$, $\beta_1$, and $\beta_2$ are fitting parameters. The total resistivity $(\rho(\mathbf{B}) = \rho_1(\mathbf{B}) + \rho_2(\mathbf{B}))$ can be expressed by $\rho(\mathbf{B}) = \frac{1}{\sigma_1(0) + \beta_1 \mathbf{B}^2} + \rho_2(0) + \beta_2 \mathbf{B}^2$ and the longitudinal magnetoresistance (LMR) equation can be expressed as $LMR(\mathbf{B}) = \frac{1}{\sigma(0) + \eta_1 \mathbf{B}^2} - \frac{1}{\sigma(0)} + \eta_2 \mathbf{B}^2$ (where $\eta_1$, $\eta_2$, and σ(0) are fitting parameters) which satisfy the condition LMR (0) = 0. This LMR equation describes our data under strain $\epsilon_1$ very well.

# Data availability

The data generated in this study and supporting the manuscript figures including those in the Supplementary Information have been deposited in the Open Science Framework database under accession code: https://osf.io/fst5d/?view_only=a8c9aa94310f400db60c7c160eb430d6

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

## Acknowledgements

This research was primarily supported by the National Science Foundation Materials Research Science and Engineering Center program through the UC Irvine Center for Complex and Active Materials (DMR-2011967). L.A.J. acknowledges the support from NSF-CAREER (DMR 2146567). M.T.P. and N.S. acknowledge support from the Laboratory Directed Research and Development program of Los Alamos National Laboratory under project number 20230014DR. This work was performed, in part, at the Center for Integrated Nanotechnologies, an Office of Science User Facility operated by the U.S. Department of Energy (DOE) Office of Science. Los Alamos National Laboratory, an affirmative action equal opportunity employer, is managed by Triad National Security, LLC for the U.S. Department of Energy's NNSA, under contract 89233218CNA000001. We acknowledge discussions of the data with Cyprian Lewandowski at Florida State University and Shunqing Shen at the University of Hong Kong. J.L and L.A.J are grateful to Jing Xia and his group at UCI for the initial exploration of using a three-piezo stack strain cell to apply strain on a $HfTe_5$ sample.

## Author contributions

J.L. and Y.Z. contribute equally to this work. L.A.J. supervised the overall research. J.L. synthesized the sample and conducted the electrical transport experiments with the help of R.A.W. Y.Z. and R.W. performed the first-principles calculations. J.L., S.Y.R., and M.A.D. designed and fabricated the strain apparatuses. T. H., M. A. D, and S. Y. R. performed the finite element analysis (FEA). Single crystal X-Ray diffraction was done by J.W.Z. ARPES measurements were performed by N.S., K.M., P.V., and N.M. J.D.S.Y. and M.T.P. gave suggestions and guidance for the experiments. J.L., L.A.J., Y.Z., R.W., and N.S. analyzed the data. All authors contributed to the interpretation of the data and to writing the manuscript.

## Competing interests

The authors declare no competing interests.
