## [Peer Review File · Nature Communications]

Reviewers' Comments:

Reviewer #1:

Remarks to the Author:

In this work, the authors report the strain-driven topological phase transition from a weak topological insulator phase to a strong topological insulator phase in HfTe5 crystal. They used the bending station apparatus to achieve high strain on the HfTe5 sample and investigated the electronic transport properties of the HfTe5 sample with a strong topological insulator state, and might observe transport signatures of the topological surface states.

However, the novelty of this work is not enough. In fact, an ARPES work and two electronic transport works have investigated the strain-driven topological phase transition of ZrTe5 [Nat. Commun. 12, 406 (2021); Sci. Adv. 5, eaav9771 (2019); Communications Materials 3, 94 (2022)]. Although, the magnitude of the strain achieved in this manuscript may be larger than those in the previous works, but the phase diagram of the whole topological phase transition of ZrTe5 has been clear through the previous experimental and theoretical studies. It is to be noted that ZrTe5 and HfTe5 have substantially the same properties. Therefore, compared with the previous works, this manuscript has not made remarkable progress in physics or experimental technology. Readers cannot gain a new understanding of ZrTe5/HfTe5 through this manuscript.

In addition, we think the linear longitudinal magnetoresistance is insufficient to identify the topological surface states. The authors probably should carry out transport measurements at a higher magnetic field and lower temperature to get nearly zero effective mass and nontrivial Berry phase from quantum oscillation or use the ARPES to observe the topological surface states under strain directly.

Since the journal Nature Communications aims at publishing high-impact and novel research that is of interest to a general reader, I cannot recommend the publication of this work. Of course, I believe that the experimental data in this manuscript are reliable. I suggest this work can be submitted to other journals.

Reviewer #2:

Remarks to the Author:

This manuscript reports extensive transport measurements together with DFT calculations, to present evidence about the in-situ strain-driven topological phase transition in transition metal pentatelluride HfTe5. Under strain, the closing and reopening of the bandgap have been observed by authors, indicating a topological transition from WTI to STI phase. Upon further increase of strain to about 4.5%, the authors claimed that a dominant TSS transport was realized in the system.

Compared with ZrTe5, the detailed studies on strain-induced evolution of electronic structure in HfTe5 seems rather limited currently. This work uses electronic transport measurement to track the topological phase transition in HfTe5, which could be seen as an important basis for other spectroscopic techniques studies such as ARPES. The results are interesting. And the data is well presented. However, some data analyses still need more discussion. Accordingly, I cannot recommend the publication of the manuscript in its current form. I have some questions and suggestions as follows and the authors should properly address:

1. As the authors mentioned, the saturation of ρ_{xx} for $T < 10$ K is due to the topological surface state dominant transport when the applied strain is up to ϵ_3 , which has been shown in Fig. 3 (d). However, it is not clear to me why this trend disappears in Fig. 3(c). Instead, the continuous increase of ρ_{xx} with decreasing temperature has been shown for the strain of ϵ_3 . Can the authors clarify this?
2. It is noticed that Figure 4 only displayed the magneto-transport results under the strain from ϵ_0 to ϵ_2 , while the corresponding results under ϵ_3 are missing in the present manuscript. Since the bulk-dominated transport at small strains and surface-state-dominated transport at high strains

are one of the important findings of this work, the analysis with the involvement of strain ϵ_3 is suggested.

3. In the manuscript, the authors argued that at high strain ϵ_2 , the 2D Fermi surface has evolved based on the extracted phase shift of $\gamma=0$ from the analysis of SdH oscillations. What about the phase shift for strain ϵ_3 ? Moreover, it would be better to test the angle (θ) dependence of the SdH oscillation frequency (F). θ is the angle between the magnetic field direction and the sample surface plane normal. If the oscillation frequency follows $1/\cos\theta$, the quantum oscillations arising from a 2D Fermi surface are strongly confirmed.

Besides, I would like to list some but not all language issues for authors to check out:

1. The sentence " As observed in the sample pasted on the piezo actuator." on page 7 is confusing.
2. The sentence " Causing the transition from the WTI phase to a Dirac semimetal phase and finally an STI phase with dominant TSS" on page 9 is not complete, which makes me a little bit confused.
3. The sentence " Te-d, Te-z, and Te-a represent Te atoms at dimer, apical, and zig-zag positions, respectively" on page 21, Te-z and Te-a should represent Te atoms at zig-zag and apical positions, respectively.

Reviewer #3:

Remarks to the Author:

What's the thickness of the crystals studied here?

When applying compressive strain on crystal, how do the authors prevent buckling?

How do the authors make sure that the strain is constant along the thickness direction? How is the crystal fixed? If it's glued at the bottom surface, how can interlayer interaction (which is weak) support the large stress throughout the layers? It is particularly hard to imagine that a strain $>2\%$ can be uniformly applied to the crystals without sliding or delamination. It helps to include data showing that the transport characteristics are reproducible upon repeated application of strain.

The claim of topological surface state from saturating low temperature resistance is a bit "handwaving". Is there any more quantitative evidence to support this claim? Otherwise the authors should not make the claim as if it is evident.

In Fig.3, the authors fit the thermal activation energy gap. Over the Lifshitz transition the Fermi level shifts through the gap with changing temperature. How did the authors fit the energy gap?

I'm not convinced by the authors argument that under large strain the magnetotransport is dominated by the surface state. Over changing strain the magneto-oscillations appear to evolve smoothly in Fig.4b. It seems to be hard to be explained by a drastic change from bulk conduction to surface conduction. For example, why aren't there 2 sets of oscillations from the bulk and the surface during such transition? Even if they co-exist in a way that is not clearly separable, there should still be some evidence in terms of the shape or amplitude of the oscillations.

Under large c-axis tensile strain, the authors' band structure calculations seem to show side energy bands across the zero energy. How do these bands affect the charge transport? For example, would that induce finite conductance at lowest temperatures and zero-Berry-phase oscillations?

First of all, we would like to thank all the referees for their time to carefully review our manuscript. We are grateful for the valuable comments and suggestions which helped us to improve our manuscript. A point-to-point response (shown in black) to all comments raised by the three referees (shown in blue) is provided below. In red we are showing the main changes we performed in the main manuscript or SI.

Response to Reviewer #1

In this work, the authors report the strain-driven topological phase transition from a weak topological insulator phase to a strong topological insulator phase in HfTe₅ crystal. They used the bending station apparatus to achieve high strain on the HfTe₅ sample and investigated the electronic transport properties of the HfTe₅ sample with a strong topological insulator state, and might observe transport signatures of the topological surface states.

Response: We thank the reviewer for this accurate summary of our manuscript. We'd like to provide some additional context to further clarify our study.

In our investigation, we initially employed a piezo stack actuator as the substrate to apply a controlled, small strain to our HfTe₅ samples. This initial step, in combination with our first-principles calculations, allowed us to confirm the weak topological insulator (WTI) nature of the as-grown HfTe₅ samples synthesized through the chemical vapor transport method. This confirmation of the WTI state served as a crucial foundation for our subsequent exploration of the topological phase transition from a WTI phase to a strong topological insulator (STI) phase.

Subsequently, to achieve high levels of strain on the HfTe₅ samples, we designed and constructed a custom-built bending strain apparatus. This unique apparatus enabled us to exert substantial strain on our samples, leading to distinctive electrical transport behaviors that are associated with the topological surface states. Notably, the use of such bending strain apparatuses for electrical transport studies represents a new aspect of our work.

We want to highlight our meticulous sample preparation process, which involved careful polishing and annealing of the titanium beam substrate to ensure effective electrical isolation of the sample. Additionally, we made a deliberate choice of epoxy to ensure a strong and reliable bond between the Ti beam and the sample, which was critical for an efficient strain transmission.

We hope this additional context provides a comprehensive understanding of the methodology and significance of our research.

However, the novelty of this work is not enough. In fact, an ARPES work and two electronic transport works have investigated the strain-driven topological phase transition of ZrTe₅ [Nat. Commun. 12, 406 (2021); Sci. Adv. 5, eaav9771 (2019); Communications Materials 3, 94 (2022)]. Although, the magnitude of the strain achieved in this manuscript may be larger than those in the previous works, but the phase diagram of the whole topological phase transition of ZrTe₅ has been clear through the previous experimental and theoretical studies. It is to be noted that ZrTe₅ and HfTe₅ have substantially the same properties. Therefore, compared with the previous works, this manuscript has not made remarkable progress in physics or experimental technology. Readers cannot gain a new understanding of ZrTe₅/HfTe₅ through this manuscript.

Response: We thank the reviewer for commenting on the state of the art of strain-driven topological phase transitions in ZrTe_5 . We cited the references mentioned by the reviewer in our manuscript. However, we maintain that the existing publications do not diminish the novelty of our work. One key distinction is the magnitude of strain applied in our study. Unlike previous works that primarily employed small strains on ZrTe_5 , our research pioneers the application of substantial strain to observe electrical signatures associated with the topological surface states in HfTe_5 (We will discuss more about these signatures below). In the cited references, the maximum strain typically reached close to 0.4%, either tensile or compressive. With such limited strain, the observed effects included gap closure through ARPES, modulation of negative linear magnetoresistance, or alterations in the bulk Shubnikov-de Haas oscillations. To unveil the transport signatures of the topological surface states and diminish the influence of bulk states, larger strains are required.

Over the past decade, since the prediction of large-gap quantum spin Hall insulator in topological pentatellurides, substantial efforts have been devoted to the study of ZrTe_5 . *We believe the physics in HfTe_5 is equally rich, if not richer, owing to its stronger spin-orbital coupling.* However, HfTe_5 has received much less attention. In the literature, one can find an equal number of experimental works asserting that as-grown ZrTe_5 is an STI versus a WTI, even for crystals grown using the same method, such as the chemical vapor transport (CVT) method. The topological state for as grown HfTe_5 samples also needed more investigation. Recently, a topological Lifshitz transition to a 1D Weyl mode [Ref1], which can only occur in the WTI scenario, was proposed to explain the results of magneto-infrared spectroscopy in HfTe_5 under high magnetic fields¹. In this manuscript, we have confirmed the WTI state in our as-grown CVT HfTe_5 samples by combining DFT calculations with transport measurements. Specifically, for HfTe_5 , a particularly promising avenue for our future investigations is tracking the development of the proposed 1D Weyl mode with strain through electrical and magneto-optical measurements. The existence of the 1D Weyl nodes will depend on our ability to manipulate the strain conditions.

Hence, our manuscript presents compelling evidence through electrical transport measurements that showcase a topological phase transition (TPT) from a WTI to an STI in our HfTe_5 samples, with a dominant contribution from topological surface states when exposed to substantial strain. We have calculated the surface states in HfTe_5 and our preliminary results show that they are different than the surface states in other STI materials, like Bi_2Se_3 . Our calculations show that the up spin and down spin mix and they are not good quantum numbers, in contrast to other STIs. But this is beyond the scope of this manuscript. Consequently, we assert that our work introduces a novel approach by demonstrating the feasibility of applying substantial strains to quantum materials. As a result, we believe our manuscript represents a significant advancement in the fields of physics and experimental technologies.

In addition, we think the linear longitudinal magnetoresistance is insufficient to identify the topological surface states.

Response: We agree with the referee that the linear magnetoresistance by itself is insufficient to identify the topological surface states. Instead, within this manuscript, we present six critical observations that collectively support the dominant topological surface state transport when a large strain is applied along the crystal c axis of HfTe_5 . **Additionally, in this revised manuscript**

we are presenting data from a new sample, B2 which was strained to a high strain of ~4.5% (similar to ϵ_3 applied to sample B1).

- 1) **Drastic Increase in Sample Resistance:** When applying strain along the crystal c axis, we observe a remarkable increase in the sample resistance—amounting to 24,200% in sample B1 with strain up to ~4.5% and 190,500% in sample B2 under a similarly large strain. These changes in resistance significantly exceed the variations of 15% and 170% reported in previous studies on ZrTe_5 . This drastic increase in resistance at low-Ts points towards a bulk gap increasing with strain along the c axis as also demonstrated by our DFT calculations. Interestingly, this insulating behavior at low-Ts and the resistance plateau mirrors observations in intrinsic 3D strong topological insulators, such as BiSbTeSe_2 , and topological Kondo insulators, such as SmB_6 , where the surface states dominate the conduction at low temperatures.
- 2) **Surface contribution of the conductivity:** Because of the resemblance of the resistance plateau to other intrinsic 3D STIs, we employed a fitting procedure to model the temperature dependence of resistance (R vs. T) under high strain. To more quantitatively extract the surface contribution to the total conductance (G^{tot}), we fit our $R_{\text{sh}}(T)$ data to a simple model used in ref. [45], where the total (sheet) conductance $G^{\text{tot}} = 1/R_{\text{sh}}(T)$ is the parallel sum of the thermally activated bulk conductance which contributes with $G^{\text{bulk}}(T) = t(\rho_{b0}e^{\Delta/kT})^{-1}$, where k is the Boltzmann constant with the fitting parameters being ρ_{b0} , the high temperature bulk resistivity, and Δ , the activation energy, and a metallic surface conductance $G^{\text{sur}}(T) = (R_{\text{sh}0} + AT)^{-1}$, where the fitting parameters are $R_{\text{sh}0}$, representing the low-T residual resistance (due to impurity scattering), and A , reflecting the electron-phonon scattering. **The fitting results consistently yield 100% surface conduction at low temperatures as now discussed in the main manuscript. Particularly for sample B2 we obtain 100% surface conduction for temperatures below 50K, while for sample B1 for temperatures below 10K.**
- 3) **Shubnikov-de Haas Oscillations:** For sample B2 at strain ~ 4.5% the resistivity has increased 190,500% compared when no strain is applied. At this high resistive state, we still observe SdH oscillations, even where our analysis suggests that transport is dominated by surface states and there is no bulk Fermi surface. Very interestingly, the SdH oscillations disappear when a magnetic field is applied in-plane (along the a axis). This could only be understood by surface conduction. **This phenomenon, not previously reported, is now more convincingly illustrated through experiments conducted at higher magnetic fields ($B = 12$ T). To illustrate this important point, as suggested by Reviewers 2 and 3, we have performed more experiments at a higher magnetic field ($B = 12$ T) and added two new panels to Figure 4 (Fig. 4 c & d) to the main text and one new figure (Fig. S16) to the Supplementary Information (SI).** In the sample with zero strain, quantum oscillations are present at any angles with frequencies well fitted by a 3D ellipsoid Fermi surface. However, under large strains, the quantum oscillations vanish when the magnetic field aligns around the in-plane direction with $\theta > 70^\circ$, where θ is the angle between the magnetic field and the sample out-of-plane direction. The quantum oscillation frequencies for $\theta < 70^\circ$ can be well fitted to a $1/\cos\theta$ function, characteristic of 2D systems. This finding reinforces the dominance of 2D surface states in transport under large strain.
- 4) **Berry Phase Analysis:** Given the small Fermi surfaces of both bulk and surface states, we note that the last (0 th) Landau level (n) can be reached at magnetic fields below 9T.

Our Berry phase analysis, based on quantum oscillations for $n > 2$ (magnetic field $B < 1$ T) to mitigate Zeeman effect influence, reveals the Berry's phase remains to be π with a dimensional factor (δ) changing from $\delta = 0.125$ (3D case) when no strain is applied to $\delta = 0$ (2D case) under significant strain (ϵ_2) for sample B1. **We have verified this finding by repeating strain measurements on a different sample, B2, with corresponding data included in the main manuscript and the SI.** This suggests that under high strain, carriers in HfTe_5 behave like 2D Dirac fermions.

- 5) **Linear Longitudinal Magnetoresistance:** The linear longitudinal MR serves as a fifth signature of a strain induced TPT with dominant surface state conduction. Similar observations have been made in other topological insulators.
- 6) **Small and tunable cyclotron mass:** Following the reviewer's suggestion, we extracted the cyclotron effective mass (m^*) of electrons in HfTe_5 at different strain levels. First, m^* is small at zero strain $\sim 0.023m_e$ (where m_e is the electron mass at rest) and it decreases with increasing strain to $\sim 0.017m_e$ for ϵ_2 . This reduction of m^* may be attributed to the linear Dirac cone dispersion of the surface state electrons.

These six observations, collectively presented in our manuscript, provide a robust foundation for our assertion of surface state domination when HfTe_5 is subjected to significant strain along the crystal c axis direction. This comprehensive evidence underscores the significance of our study in advancing the understanding of strain-induced topological phase transitions in solid state materials.

The authors probably should carry out transport measurements at a higher magnetic field and lower temperature to get nearly zero effective mass and nontrivial Berry phase from quantum oscillation or use the ARPES to observe the topological surface states under strain directly.

Response: We have conducted high magnetic field transport measurements up to 60 T and temperatures in the 300mK range for samples without strain for another project. However, applying strain at these elevated magnetic fields presents a significant technical challenge and is an ongoing aspect of our research. It is noteworthy that one intriguing avenue of investigation is the comparison of the quantum limit behaviors between the STI and WTI states. Nevertheless, this endeavor falls beyond the scope of this work.

Thanks to the reviewer's comment **we have added the data of SdH oscillations measured at various temperatures, and the cyclotron effective mass (m^*) extraction in the SI, Fig. S14.** m^* is nearly zero at zero strain, which is consistent with previous reports. With increased strain, m^* tends to become smaller. At ϵ_2 , we see a significant reduction ($\sim 22\%$) in effective mass, which may be attributed to the linear Dirac cone dispersion of the surface electrons. However, we acknowledge that this may not exclusively result from TSS dominant conduction. **We added this discussion in the SI.**

Concurrently, our collaborators at LANL are conducting ARPES measurements on samples subjected to strain. It's noteworthy that in reference [33], ARPES measurements were conducted while applying strain to a ZrTe_5 sample. In their measurements, they exclusively applied strain along the a axis, resulting in an increased bulk gap under tensile strain. They also applied a small compressive strain, which enabled them to observe the closure of the bulk gap. However, they encountered issues with sample buckling when attempting to apply significant compressive strain ($> 0.5\%$ strain). It's crucial to emphasize that this specific paper solely focused on ARPES measurements, and no electrical transport experiments were performed.

Furthermore, their study was limited in terms of the amount of compressive strain they could apply to achieve an STI phase.

It was this ARPES study that initially inspired our current research, as it highlighted the substantial tunability of the bulk gap with strain. To reach the STI phase and to avoid sample buckling under compressive strain, we have applied large tensile strain along the c axis. Within the confines of our manuscript, we provide evidence through electrical transport measurements that substantial strain results in the dominance of topological surface states in transport. While we plan to collaborate with our partners to visualize the surface states in ARPES experiments, this aspect falls beyond the current scope of our work.

Since the journal Nature Communications aims at publishing high-impact and novel research that is of interest to a general reader, I cannot recommend the publication of this work. Of course, I believe that the experimental data in this manuscript are reliable. I suggest this work can be submitted to other journals.

Response: We hope to have successfully addressed the Reviewer's concerns, making our work suitable for publication in Nature Communications, as indicated in our earlier responses. Additionally, we would like to highlight several other significant and innovative contributions from our study:

- 1) Bending Strain Apparatus: Our adaptation of the bending strain apparatus for electrical transport measurements presents a versatile tool that can be extended to conduct extensive strain engineering experiments on various bulk van der Waals materials or low-dimensional van der Waals materials and heterostructures. In line with the reviewer's suggestion, for future projects and follow-up investigations, the bending strain apparatus can serve as a platform not only for electrical measurements under strain but also for a range of dependent measurements, encompassing electronic-thermal, optical, and mechanical assessments under extreme environmental conditions. This includes high magnetic fields (either DC or pulsed magnets) and ultra-low temperatures achievable in a dilution refrigerator. The compact size of the apparatus, the absence of high voltage requirements, and the efficient thermal contact between the sample and the metal base make it an adaptable and invaluable tool.
- 2) DFT calculations: Our first-principles calculation study represents a pioneering application of the SCAN exchange-correlation functional in conjunction with rVV10 van der Waals corrections to treat the pentatelluride systems. The SCAN meta-generalized gradient approximation demonstrates its capability to accurately model short- to intermediate-range vdW interactions. We conclude that SCAN with rVV10 vdW correction offers a superior description of the TPT in pentatellurides featuring Te-Te bonds within the range of 2.7 - 4 Å. The WTI state for pristine pentatellurides was not well perceived by previous DFT calculations. Our results could help to clarify the debate of STI or WTI remaining in literature.
- 3) Electronic Transport Properties: Our study reveals the remarkable transformation of electronic transport properties in HfTe_5 in response to strain. This finding not only enriches our understanding of HfTe_5 but also provides inspiration for future investigations into strain-dependent experiments. These may include probing the electronic structure and exploring electronic transport phenomena such as the nonlinear Hall effect across the topological phase transition in HfTe_5 .

Reviewer #2 (Remarks to the Author):

This manuscript reports extensive transport measurements together with DFT calculations, to present evidence about the in-situ strain-driven topological phase transition in transition metal pentatelluride HfTe₅. Under strain, the closing and reopening of the bandgap have been observed by authors, indicating a topological transition from WTI to STI phase. Upon further increase of strain to about 4.5%, the authors claimed that a dominant TSS transport was realized in the system.

Compared with ZrTe₅, the detailed studies on strain-induced evolution of electronic structure in HfTe₅ seems rather limited currently. This work uses electronic transport measurement to track the topological phase transition in HfTe₅, which could be seen as an important basis for other spectroscopic techniques studies such as ARPES. The results are interesting. And the data is well presented. However, some data analyses still need more discussion. Accordingly, I cannot recommend the publication of the manuscript in its current form. I have some questions and suggestions as follows and the authors should properly address:

Response: We thank the Reviewer for considering our results as interesting.

1. As the authors mentioned, the saturation of ρ_{xx} for $T < 10$ K is due to the topological surface state dominant transport when the applied strain is up to ϵ_3 , which has been shown in Fig. 3 (d). However, it is not clear to me why this trend disappears in Fig. 3(c). Instead, the continuous increase of ρ_{xx} with decreasing temperature has been shown for the strain of ϵ_3 . Can the authors clarify this?

Response: We would like to provide clarification regarding the data presented in Fig. 3c and Fig. 3d. It's important to note that the data shown in Fig. 3c is identical to that displayed in Fig. 3d, with the only distinction being the choice of scale. Specifically, Fig. 3c employs a linear-linear scale, whereas Fig. 3d employs a log-log scale. We have addressed this distinction in both the main text and the figure caption to eliminate any potential confusion.

In the main text, we have included an explanation of this difference: "The temperature dependence of ρ_{xx} of sample B1 shows a remarkable evolution with strain as seen in Fig. 3c (for clarity Fig. 3d is plotted on a log-log scale)."

Additionally, we have rephrased the figure captions to enhance clarity: "**d**, ρ_{xx} vs. T for B1 at different strains and B2 at $\epsilon \sim 4.5$ % plotted in linear-linear scale in c and plotted in log-log scale in d, for clarity."

Furthermore, we have conducted measurements on another sample (sample B2) subjected to significant strain ($\sim 4.5\%$) and have observed a saturation of ρ_{xx} even at temperatures below 50K. This continuous increase in ρ_{xx} when plotted in linear-linear scale, coupled with the saturation observed in log-log scale as temperature decreases, collectively suggest the presence of surface state conduction in conjunction with semiconducting bulk transport. **A new figure is included in the main text, Figure 3f and in the SI, Figure S15.**

2. It is noticed that Figure 4 only displayed the magneto-transport results under the strain from

ϵ_0 to ϵ_2 , while the corresponding results under ϵ_3 are missing in the present manuscript. Since the bulk-dominated transport at small strains and surface-state-dominated transport at high strains are one of the important findings of this work, the analysis with the involvement of strain ϵ_3 is suggested.

Response: This insightful observation from the reviewer warrants further clarification. It is important to note that during the application of ϵ_3 , the sample broke when we started to measure the magnetic field dependence of the resistance at the base temperature. Consequently, we were limited to conducting resistance vs. temperature (R vs. T) measurements, and we were unable to investigate the magneto-transport measurements for ϵ_3 . However, it's crucial to emphasize that even at ϵ_2 , which is already a substantial level of strain, we were able to detect significant changes in the quantum oscillation phase with a Berry phase of π and a dimensional factor of 0, indicating a transition to a 2D Fermi surface, as well as an increase in resistance at low temperatures. Furthermore, to establish the reproducibility of our observations, we conducted measurements on an additional sample (B2) subject to $\sim 4.5\%$ tensile strain along the c axis, which demonstrates a larger increase in ρ_{xx} at low temperatures with a clearer saturation of ρ_{xx} for $T < 50\text{K}$, as shown in Figure S15.

3. In the manuscript, the authors argued that at high strain ϵ_2 , the 2D Fermi surface has evolved based on the extracted phase shift of $\gamma=0$ from the analysis of SdH oscillations. What about the phase shift for strain ϵ_3 ? Moreover, it would be better to test the angle (θ) dependence of the SdH oscillation frequency (F). θ is the angle between the magnetic field direction and the sample surface plane normal. If the oscillation frequency follows $1/\cos\theta$, the quantum oscillations arising from a 2D Fermi surface are strongly confirmed.

Response: Indeed, conducting magneto-transport measurements at the highest strain level for sample B1 would have been an ideal approach. However, as mentioned above, our sample broke at low temperatures under strain ϵ_3 .

In response to the valuable suggestion from the reviewer, we have carried out measurements of the angular dependence of the Shubnikov-de Haas (SdH) oscillations in another sample (B2) under a similar high strain condition $\sim 4.5\%$ as ϵ_3 . Our observations reveal that a phase shift $\gamma = 0$ is obtained, distinct from $\gamma = 0.11$ at the zero-strain case, as shown in the newly included Fig. S15.

Additionally, the SdH oscillations are detectable at all angles when no intentional strain is applied, consistent with the expectation of a 3D Fermi surface. However, under a high strain with the sample showing bulk insulating behavior, we can only detect SdH oscillations for $\theta < 70^\circ$ (θ is depicted in the inset of Figure S16a). For $\theta > 70^\circ$, when the magnetic field is applied around the in-plane direction, we no longer observe SdH oscillations. This phenomenon is demonstrated in sample B2, and also in sample B1 (the original bending sample in the main text) for $\theta = 90^\circ$. We have incorporated a new figure in the SI Fig S16. There, we show the ΔR_{xx} vs $1/B$ at different angles for an as grown sample (F2 with no strain applied) depicted in Figure S16a and a strained sample (B2 with strain $\sim 4.5\%$) as shown in Fig. S16b. This measurement further confirms that the Fermi surface of the highly strained HfTe_5 sample becomes two dimensional in nature.

Besides, I would like to list some but not all language issues for authors to check out:

1. The sentence " As observed in the sample pasted on the piezo actuator." on page 7 is confusing.
2. The sentence " Causing the transition from the WTI phase to a Dirac semimetal phase and finally an STI phase with dominant TSS" on page 9 is not complete, which makes me a little bit confused.
3. The sentence " Te-d, Te-z, and Te-a represent Te atoms at dimer, apical, and zig-zag positions, respectively" on page 21, Te-z and Te-a should represent Te atoms at zig-zag and apical positions, respectively.

Response: we appreciate the reviewer's comments and we have corrected the language in our manuscript.

1. The sentence has been rephrased to: "While below 20 K we observe a distinct temperature dependence compared to that of free-standing samples, we observe ρ_{xx} increases with decreasing T. At $T = 20$ K, ρ_{xx} shows a minimum, marking the closure of the band gap. For $T < 20$ K, the band gap increases or reopens by ϵ_1 and low temperatures. A similar trend of resistivity upturn for $T < 30$ K was observed in sample P1 (Supplementary Fig. S10a), which experiences an effective strain of $\sim 0.475\%$ along the c axis at base temperature (See detailed discussions in Section VIII of the SI).".
2. The sentence has been rephrased as: Once the gap reopens, the TSSs are formed. In other words, increasing strain causes a topological phase transition from the WTI phase to a Dirac semimetal phase, and finally to an STI phase with non-trivial surface states contributing to the electronic transport.
3. We thank the reviewer for noticing this mistake. The sentence has been corrected as " Te-d, Te-z, and Te-a represent Te atoms at dimer, zig-zag, and apical positions, respectively".
4. We carefully corrected other places where we find that the language may have been misleading.

Reviewer #3 (Remarks to the Author):

What's the thickness of the crystals studied here?

Response: The thickness of our transport measurement samples ranges from 50 to 110 μm , specifically 55 ± 2 μm for sample B1, and 67 ± 2 μm for sample B2. We have added this information in the paragraph of "Transport measurements" in the Methods section.

When applying compressive strain on crystal, how do the authors prevent buckling?

Response: Addressing the issue of buckling in our experiments is of paramount importance when applying compressive strain to crystals, particularly in the context of van der Waals materials. We initially encountered this challenge when attempting to suspend HfTe_5 samples across a gap using a three piezo-stack strain cell, as illustrated in Fig. R1. Unfortunately, this setup resulted in unavoidable buckling of our samples, rendering the data collected with it unsuitable for inclusion in our current manuscript. Consequently, we reconfigured our experimental approach.

Figure R1. Application of strain on HfTe_5 with a three piezo-stack strain cell. **a.** Schematic of experimental setup of a three piezo-stack strain cell. **b.** An optical image of a sample under a compressive strain $<1\%$. The thick red arrow points to the slight bow shape and the kink found on the sample. The scale bar represents 1 mm.

In the data presented in this manuscript, we exclusively applied compressive strain when affixing samples P1 and P2 to a piezo stack actuator. In this setup, the maximum compressive strain reached was 8×10^{-4} . With such a low level of compressive strain and the secure attachment of the sample to the substrate, we did not observe any instances of buckling in the samples.

However, it's essential to note that achieving a substantial strain is necessary to induce a topological phase transition, and to observe the surface states dominant transport. To overcome the issue of buckling, we opted to apply tensile strain along the c axis (the shorter in-plane direction of the sample) instead of compressive strain along the a axis (the longer direction of the sample). To achieve this, we oriented the elongated rectangular sample perpendicular to the length direction of the Ti beam's top surface, as depicted in Figure 3a and Figure 3b i and ii. Due to the similarity in Poisson ratios between the Ti beam and the sample, the sample experienced a compressive strain along the a axis as well. Crucially, this arrangement prevented the sample from buckling under substantial tensile strain. It's worth mentioning that we attempted to apply compressive strain along the a axis to another sample by adhering it to the bottom of the Ti beam. However, this approach led to the sample buckling under strains exceeding 0.3%.

Furthermore, to highlight this setup, we have rephrased and incorporated the following sentence in the manuscript on page 6: "Our measurement setup can prevent the sample from buckling, as the sample is affixed at the center of the Ti beam's top surface with uniform strain distribution and we exclusively apply tensile strain along the c axis when using the "bending station", as

shown in Fig. 3a.” instead of “This setup can prevent the sample from buckling as the sample is glued at the center of the Ti beam top surface.” as before.

We hope that these clarifications and additions enhance the understanding of our experimental methodology for the benefit of the readers.

How do the authors make sure that the strain is constant along the thickness direction? How is the crystal fixed? If it's glued at the bottom surface, how can interlayer interaction (which is weak) support the large stress throughout the layers? It is particularly hard to imagine that a strain >2% can be uniformly applied to the crystals without sliding or delamination. It helps to include data showing that the transport characteristics are reproducible upon repeated application of strain.

Response: We thank the reviewer for their insightful questions regarding the intricacies of strain application. We concur with the reviewer's concern that the weak interlayer interactions inherent in van der Waals materials might not suffice to uniformly transmit substantial strain across the layers.

In our efforts to ensure a consistent and robust bond between the sample's bottom surface and the top surface of the Ti substrate beam, we have employed epoxy with a lower viscosity. When placing the samples on top of the epoxy layer, the epoxy will go on to the side surfaces of the samples. Once the epoxy is cured, it takes on a structural form as simplified in Fig. S12b.

Motivated by the reviewer's comment **we conducted additional simulations using COMSOL. These simulations have confirmed that such a structural arrangement effectively facilitates the uniform application of significant stress along the interlayer direction. We have included the results of these COMSOL simulations in the SI.**

Furthermore, we have acknowledged and credited a co-author who contributed to the COMSOL simulations in recognition of their valuable assistance in this aspect of our research.

The claim of topological surface state from saturating low temperature resistance is a bit “handwaving”. Is there any more quantitative evidence to support this claim? Otherwise the authors should not make the claim as if it is evident.

Response: We wish to extend our gratitude to the reviewer for their valuable comments and constructive suggestions. In response to the reviewer's feedback, we have made significant revisions to our manuscript.

The saturation of low-temperature resistance in strained HfTe₅ samples exhibits transport behavior reminiscent of well-known topological insulators with topological surface state conduction. To provide a more comprehensive analysis, as shown in Fig. 3f we fit our $R_{sh}(T)$ data to a simple model used in ref. [45], where the total (sheet) conductance $G^{tot} = 1/R_{sh}(T)$ is the parallel sum of the thermally activated bulk conductance which contributes with $G^{bulk}(T) = t(\rho_{b0}e^{\Delta/kT})^{-1}$, where t is the thickness of the sample, k is the Boltzmann constant with the fitting parameters being ρ_{b0} , the high temperature bulk resistivity, and Δ , the activation energy, and a metallic surface conductance $G^{sur}(T) = (R_{sh0} + AT)^{-1}$, where the fitting parameters are R_{sh0} , representing the low-T residual resistance (due to impurity scattering), and A , reflecting the electron-phonon scattering. The determination of the surface fitting parameter becomes feasible when the resistance vs. temperature saturates. For our samples, B1 exhibits a well-fitted region

for $T < 10\text{K}$, while B2 shows good fitting for $T < 50\text{K}$. Consequently, once we obtain the values of $R_{\text{sh}0}$ and A , we plot the ratio of the surface conductance contribution to total conductance. Our revised Fig. 3f now illustrates $G^{\text{sur}}/G^{\text{tot}}$ for samples B1 and B2. It is noteworthy that surface transport dominates for sample B1 at $T < 10\text{K}$ and for sample B2 when $T < 50\text{K}$. At higher temperatures, the surface contribution diminishes, and a strong bulk contribution emerges at room temperature.

These observations suggest that, at low temperatures and under high strain conditions, the dominant conduction mechanism in our samples is associated with surface states.

Moreover, our magneto-transport measurements suggest that these surface states are 2D in nature, as evidenced by the well-fitted Shubnikov-de Haas oscillations with a $1/\cos\theta$ dependence. The dimensional factor of 0 together with the π Berry phase obtained at high strain further implies that these 2D surface states possess topological characteristics. Furthermore, our DFT simulations indicate that the application of significant strain along the c axis results in the opening of a larger bulk gap, with increasing topological surface states filling the bulk gap and reducing bulk conductivity. All these transport signatures collectively suggest that our samples exhibit characteristics of a strong topological insulator phase with dominant surface state transport.

In response to the reviewer's suggestion, we have incorporated our surface-to-total conductivity analysis into Figure 3f and provided further details in the SI. These additions enhance the comprehensibility and robustness of our research findings.

We deeply appreciate the reviewer's insightful input, which has significantly contributed to the clarity and accuracy of our work, and we believe that these enhancements will facilitate readers' comprehension of our study.

In Fig.3, the authors fit the thermal activation energy gap. Over the Lifshitz transition the Fermi level shifts through the gap with changing temperature. How did the authors fit the energy gap?

Response: We appreciate the reviewer's question concerning our estimation of the thermal activation energy. Figure 3e presents the plot of $\log(\rho_{xx})$ vs. $1/T$ at various strain levels. Our estimation of the thermal activation energy was carried out for temperatures above 70K and in small temperature ranges, a choice made to mitigate the potential influence of the Lifshitz transition. This decision aligns with previous ARPES measurements, which did not observe significant gap changes at high temperatures Ref. [46].

In order to provide clarity on this point, we have introduced the following sentence into the main manuscript: "We selected temperatures exceeding 70K to ensure that the estimate of the thermal activation gap remains unaffected by the Lifshitz transition."

I'm not convinced by the authors argument that under large strain the magnetotransport is dominated by the surface state. Over changing strain the magneto-oscillations appear to evolve smoothly in Fig.4b. It seems to be hard to be explained by a drastic change from bulk conduction to surface conduction. For example, why aren't there 2 sets of oscillations from the bulk and the surface during such transition? Even if they co-exist in a way that is not clearly

separable, there should still be some evidence in terms of the shape or amplitude of the oscillations.

Response: We appreciate the reviewer's question, and we are confident in our assertion that the topological surface state dominates transport at high strains, as detailed in our previous discussion. To gain a better understanding of the magneto-transport contributed by the topological surface states, we conducted band structure calculations for the surface states at the top and bottom of the sample, as illustrated in Fig. R2. The band dispersions of the surface states and the bulk state appear to be very closely aligned.

Figure R2. The topological surface states spectrum of the top surface of HfTe₅ with increased strain along the c axis of HfTe₅ **a-c**. Note: a trend of the Fermi energy shifting toward the valence band is observed, but the exact energy shift may not be accurate.

Given that the bulk Fermi level is in close proximity to the valence band edge in the as-grown sample, applying a magnetic field perpendicular to the sample's ac plane at high strain levels should result in a Fermi surface cross-section of the topological surface states that closely resembles that of the bulk state. This is why we believe there is no significant alteration in the quantum oscillation (QO) frequency when the magnetic field is perpendicular to the sample's ac plane. However, the differences in dimensionality between the two Fermi surfaces do lead to noticeable consequences, including changes in the phase factor of QOs and the angular dependence of the QOs.

In all of our data sets, we have not observed any clear signatures of two distinct sets of oscillations at any strain level, as depicted in Figure R3. There is no evidence of two separate sets of oscillations, one originating from the bulk and the other from the surface. This absence of differentiation may be attributed to the similar electron densities associated with both the surface states and the bulk states. The bulk states exhibit a low density and possess a small effective mass, approximately $\sim 0.02 m_0$. Given the diminutive mass of the bulk and the already small effective mass of the surface states, their characteristics overlap. It is important to note that distinct sets of oscillations might become discernible if a gate voltage were applied. However, this approach is not feasible with bulk samples. We are currently exploring thinner samples and investigating the potential use of a gate voltage to modulate the chemical potential. Nevertheless, these investigations fall outside the scope of the present paper.

To provide additional clarity on this matter, **we have included the following statement in the SI: "We would like to note that we have not observed an abrupt change in the SdH oscillations with strain. This may be due to the small and similar sizes of the Fermi surfaces from the bulk and surface states, as suggested by our DFT calculations and as indicated by our measured small**

oscillating frequency of ~ 1 T and the light cyclotron effective masses measured at all strain levels. Also, we have not observed any beating patterns in the quantum oscillations or other signatures of the coexistence of bulk and surface states carriers. This may be due to the 100% surface contribution we observed in the ρ_{xx} vs. T with an insulating bulk at low temperatures and high strains."

Under large c-axis tensile strain, the authors' band structure calculations seem to show side energy bands across the zero energy. How do these bands affect the charge transport? For example, would that induce finite conductance at lowest temperatures and zero-Berry-phase oscillations?

Response: We are confident that the side bands calculated theoretically have a minimal or negligible impact on charge transport when compared to the topological surface states (TSS).

Figure R3. Fast Fourier Transform (FFT) of the Shubnikov-de Hass oscillations for sample B1 under different strains (a-c) and sample B2 under strain $\sim 4.5\%$.

This conviction is rooted in the six reasons previously discussed that favor dominant surface transport.

The key observation supporting this claim is the saturation of resistance in sample B2 when subjected to a strain of approximately 4.5%. This observation contradicts the scenario of a side band contributing to transport because if a side band would be intersecting the Fermi level, we would expect a reduction in resistivity, not a saturation. However, such a reduction has not been observed in our samples. Instead, we consistently observe resistance saturation, aligning well with the notion of dominant topological surface state transport.

Furthermore, our Shubnikov-de Haas oscillations measurements have not revealed any additional frequencies, indicating the absence of other high-mobility bands influencing transport. Based on these experimental findings, we have found no evidence supporting the idea that the side band significantly affects the transport measurements in our samples.

To provide further clarification on this matter, **we have included the following statement in the SI: "Our simulations show that with increased strain a side band becomes important and could contribute with extra carriers, however we have not found experimental evidence of this extra band in our measurements. This may be due to a large effective mass band and potentially lower mobility."**

Ref1. Wu, W. *et al.* Topological Lifshitz transition and one-dimensional Weyl mode in HfTe₅. *Nat. Mater.*

22, 84–91 (2023).

Reviewers' Comments:

Reviewer #1:

Remarks to the Author:

I think the authors have carefully replied to my comments. They have done more experiments and calculations.

Reviewer #2:

Remarks to the Author:

The authors have addressed all my concerns both in the point-to-point responses and in the revised manuscript. Moreover, the manuscript has been greatly improved at present. Thus, I support the publication of the manuscript in Nature Communications.

Reviewer #3:

Remarks to the Author:

I'm happy with the replies from the authors. I feel that the manuscript can be published in Nature Communication in its current form.